# Liposomal Rifabutin—A Promising Antibiotic Repurposing Strategy against Methicillin-Resistant *Staphylococcus aureus* Infections

**DOI:** 10.3390/ph17040470

**Published:** 2024-04-08

**Authors:** Jacinta O. Pinho, Magda Ferreira, Mariana Coelho, Sandra N. Pinto, Sandra I. Aguiar, Maria Manuela Gaspar

**Affiliations:** 1Research Institute for Medicines (iMed.ULisboa), Faculty of Pharmacy, Universidade de Lisboa, Av. Prof. Gama Pinto, 1649-003 Lisboa, Portugal; pinho.jacinta@campus.ul.pt (J.O.P.); magda.ferreira@campus.ul.pt (M.F.); mariana.coelho@ff.ulisboa.pt (M.C.); 2Center for Interdisciplinary Research in Animal Health (CIISA), Faculty of Veterinary Medicine, Universidade de Lisboa, 1300-477 Lisboa, Portugal; siraguiar@gmail.com; 3Associate Laboratory for Animal and Veterinary Sciences (AL4AnimalS), Faculty of Veterinary Medicine, Universidade de Lisboa, 1300-477 Lisboa, Portugal; 4iBB-Institute for Bioengineering and Biosciences and Associate Laboratory i4HB−Institute for Health and Bioeconomy at Department of Bioengineering, Instituto SuperiorTécnico, Universidade de Lisboa, 1049-001 Lisboa, Portugal; sandrapinto@ist.utl.pt; 5IBEB, Institute of Biophysics and Biomedical Engineering, Faculty of Sciences, Universidade de Lisboa, Campo Grande, 1749-016 Lisboa, Portugal

**Keywords:** *Staphylococcus aureus* infection, methicillin-resistant bacteria, clinical isolates, planktonic bacteria, biofilm, rifabutin, liposomes, therapeutic strategy

## Abstract

Methicillin-resistant *Staphylococcus aureus* (M RSA) infections, in particular biofilm-organized bacteria, remain a clinical challenge and a serious health problem. Rifabutin (RFB), an antibiotic of the rifamycins class, has shown in previous work excellent anti-staphylococcal activity. Here, we proposed to load RFB in liposomes aiming to promote the accumulation of RFB at infected sites and consequently enhance the therapeutic potency. Two clinical isolates of MRSA, MRSA-C1 and MRSA-C2, were used to test the developed formulations, as well as the positive control, vancomycin (VCM). RFB in free and liposomal forms displayed high antibacterial activity, with similar potency between tested formulations. In MRSA-C1, minimal inhibitory concentrations (MIC) for Free RFB and liposomal RFB were 0.009 and 0.013 μg/mL, respectively. Minimum biofilm inhibitory concentrations able to inhibit 50% biofilm growth (MBIC_50_) for Free RFB and liposomal RFB against MRSA-C1 were 0.012 and 0.008 μg/mL, respectively. Confocal microscopy studies demonstrated the rapid internalization of unloaded and RFB-loaded liposomes in the bacterial biofilm matrix. In murine models of systemic MRSA-C1 infection, Balb/c mice were treated with RFB formulations and VCM at 20 and 40 mg/kg of body weight, respectively. The in vivo results demonstrated a significant reduction in bacterial burden and growth index in major organs of mice treated with RFB formulations, as compared to Control and VCM (positive control) groups. Furthermore, the VCM therapeutic dose was two fold higher than the one used for RFB formulations, reinforcing the therapeutic potency of the proposed strategy. In addition, RFB formulations were the only formulations associated with 100% survival. Globally, this study emphasizes the potential of RFB nanoformulations as an effective and safe approach against MRSA infections.

## 1. Introduction

The Gram-positive bacteria *Staphylococcus aureus* (*S*. *aureus*) is a major human pathogen that can cause a wide variety of infections, from mild to life-threatening clinical conditions, including bacteremia, endocarditis, chronic osteomyelitis, and pneumonia [1,2,3]. Bacteremia refers to the presence of viable bacteria in the bloodstream, frequently arising from localized infections associated with implants or catheters [4,5]. Despite the considerable improvements in therapy, the mortality rates of *S*. *aureus* bacteremia are still high [6], with more than one in four patients perishing within 3 months [7]. The treatment of these infections is hampered by the emergence and widespread dissemination of antibiotic-resistant strains, namely methicillin-resistant *S. aureus* (MRSA) [3,8,9]. The gold standard for the clinical management of MRSA infections is vancomycin (VCM), a glycopeptide antibiotic that was approved for human use in 1958 [1,10,11]. VCM shows activity against most Gram-positive cocci and bacilli. It acts by interrupting cell wall synthesis in dividing bacteria by specifically inhibiting the incorporation of murein monomers into the peptidoglycan chain [11]. However, VCM application may be hindered by severe toxicity, low tissue penetration, and slow antibacterial effect [12]. In addition, resistance to VCM has been reported in clinical isolates of *S*. *aureus* [8,13].

Bacteria biofilms constitute another medical concern since the antibiotic penetration is weakened and the immune system action is prevented [14,15,16]. Biofilm-organized *S*. *aureus* is surrounded by a protective and complex matrix containing proteins, polysaccharides, and genomic material. In biofilm form, bacteria attach to host tissues or implanted devices, namely prosthetic joints and catheters, causing persistent infections that are resistant to treatment [16,17,18].

These challenges have prompted the search for novel and more effective therapeutic approaches. On the one hand, this may be accomplished by the discovery of new antibacterial drugs, a time- and cost-inefficient process. On the other hand, drug repurposing using nanotechnology has been accomplished mainly through the association of clinically approved antibiotics within delivery systems, namely liposomes [19,20,21,22,23]. Compared to the process of drug discovery and development, the repurposing of existing antibiotics entails shorter timelines and fewer costs [19,20]. 

Rifamycins are a class of antibiotics discovered in the 1950s and are highlighted for their activity against mycobacterial infections, comprising the clinically approved rifampicin, rifapentine, rifaximin, and rifabutin (RFB) [24,25]. Rifamycins are active against mycobacterium, Gram-positive bacteria, and, to a lower extent, Gram-negative bacteria. These antibiotics bind to the prokaryotic DNA-dependent RNA polymerase, suppressing transcription and protein synthesis [24,25,26]. As this mechanism of action is independent of bacterial division, bacteria populations with low metabolic activity, such as those in biofilms, are also susceptible [25]. Among rifamycins, RFB has recently received special attention for its potential anti-staphylococcal activity in both planktonic and biofilm bacteria [20,24,25,27]. Compared to rifampicin, RFB has lower toxicity and longer half-life, as well as weaker induction of CYP450 enzymes that results in reduced drug–drug interactions [24,26]. Moreover, RFB displays higher tissue distribution and better intracellular uptake than rifampicin, probably due to its higher lipophilicity [24]. As aforementioned, the use of lipid-based nanosystems for antibiotic delivery, namely RFB, has proven to be quite advantageous, providing protection from premature degradation and/or elimination, promoting accumulation at infected sites, and enhancing the therapeutic effectiveness [28,29]. In the literature, several examples demonstrate the advantages of using liposomes as drug delivery systems against planktonic and biofilm *S*. *aureus* (reviewed in [30,31,32,33]). Considering the potential of RFB against *S*. *aureus* infections, here the aim was to develop RFB-loaded liposomes and assess the antibacterial potency against planktonic and biofilm MRSA strains of clinical origin. Furthermore, the in vivo therapeutic effect of developed formulations was confirmed in systemic MRSA infection models (Figure 1).

## 2. Results and Discussion

### 2.1. Antibacterial Activity of Free RFB and VCM against Planktonic S. aureus

Previous studies in planktonic and biofilm methicillin-sensitive *S*. *aureus* (MSSA) (ATCC^®^25923^TM^) demonstrated the superior antibacterial potency of RFB formulations, compared to the gold-standard VCM [20]. In the present work, the antibacterial activity of RFB formulations was validated against MRSA clinical strains. First, susceptibility tests to RFB and VCM in the free form were performed in planktonic and biofilm clinical isolates, hereafter designated as MRSA-C1 and MRSA-C2. In planktonic bacteria, free antibiotics were incubated for 24 h and antibacterial activity was determined through the broth microdilution method. In Figure 2, the turbidity readings (OD_570 nm_) are depicted, and Table 1 shows the obtained MIC values.

Previously, these two antibiotics were tested against an MSSA strain, showing MIC values of 0.006 and 1.562 μg/mL for RFB and VCM, respectively [20]. Here, a similar antibacterial effect was obtained for RFB in the free form against MRSA-C1 and MRSA-C2, with MIC values of 0.009 and 0.012 μg/mL, respectively (Figure 2 and Table 1). Of note, compared to the positive control VCM, RFB was 136- and 156-fold more potent towards MRSA-C1 and MRSA-C2, respectively. 

The results obtained by turbidity measurement for MRSA-C1 were confirmed by CFU counts and by the MTT assay (Figure 3a,b). The visualization of color formation in the MTT assay is an established methodology to determine MIC values in different bacteria species, including *S*. *aureus* [34,35,36,37,38,39]. The yellow tetrazolium salt is irreversibly reduced to purple formazan crystals by metabolic active microbial cells. Here, the MIC value was defined as the minimum antibiotic concentration corresponding to the absence of color formation, compared to the negative control. MTT assay and CFU counting were also employed for planktonic MRSA-C1 incubated with VCM.

Similar MIC values were obtained among all three methods—turbidity, MTT assay, and CFU counts. For RFB and VCM in MRSA-C1, the results obtained through the visualization of formazan coloration (Figure 3a) were in accordance with those of turbidity measurements (Figure 2 and Table 1), validating this method for MIC determination. The RFB potency towards the clinical strain MRSA-C1 was further confirmed by CFU counts, as depicted in Figure 3b. At 0.026 μg/mL, RFB induced a 6 log reduction in viable bacteria in relation to Control. In comparison, VCM at an 18 fold higher concentration than RFB displayed negligible antibacterial activity.

### 2.2. Antibacterial Activity of Free RFB and VCM against Biofilm S. aureus

Following the studies in planktonic MRSA clinical isolates, the biofilm susceptibility to RFB and VCM in the free form was evaluated by MTT and crystal violet (CV) assays (Figure 4) that provide information of bacteria viability and biofilm biomass, respectively. These two assays are simple and with high reproducibility. However, it is not possible to establish a direct association between bacteria viability and biofilm biomass since CV is a nonspecific cationic dye that stains the biofilm matrix constituents and both viable and dead bacteria [40,41,42]. In Table 1, the values of MBIC_50_ for tested antibiotics in the free form against the different *S*. *aureus* strains are depicted. 

Biofilm-forming bacteria produces an extracellular matrix rich in polysaccharides, proteins, and extracellular DNA that acts as a defense against stress conditions, promoting a faster adaptation to environmental changes and increasing bacterial virulence [14,15,16]. Moreover, this complex barrier hinders the access of immune cells and antibiotics to bacteria residing within the biofilm structure, impairing the treatment effectiveness and increasing the persistence of bacterial infections [14,15,16]. It is well documented that established biofilm infections are, in general, more resistant to antimicrobial agents than free-living bacteria [20,43,44]. Nevertheless, several reports on the antibacterial effect of distinct compounds against MRSA have found MBIC_50_ values similar [45] or even inferior to MIC [46,47,48,49]. 

In the present work, RFB was equally effective against planktonic and biofilm MRSA, presenting MBIC_50_ values of 0.010 and 0.012 μg/mL for MRSA-C1 and MRSA-C2, respectively (Table 1). Due to this similar antibacterial effect, following studies were only performed with MRSA-C1. In turn, the positive control VCM did not exert an antibiofilm effect, even at the maximum tested concentration of 800 μg/mL. This lack of efficacy of VCM against *S*. *aureus* biofilm is in agreement with previous studies in MSSA (MBIC_50_ > 200 μg/mL) [20]. Also, exposure of MRSA to VCM has been reported to promote biofilm formation as a stress response [50,51]. Of note, it has been described that high concentrations of antibacterial drugs may promote bacteria survival and proliferation (reviewed in [52]). This effect has been observed for different classes of antibiotics (e.g., β-lactams, glycopeptides, quinolones, and aminoglycosides), in the presence of different microorganisms (e.g., *Staphylococcus* spp., *Streptococcus* spp., *Enterococcus* spp., *Mycobacterium* spp., and Gram-negative bacteria) (reviewed in [52]). 

### 2.3. Physicochemical Characterization of RFB-Loaded Liposomes

The current lack of specificity of antimicrobial agents and the high doses often required for a significant therapeutic outcome can cause severe toxicity and lead to drug resistance [53]. This obstacle can be overcome by associating drugs to appropriate drug delivery systems that may promote a preferential targeting to infected sites of loaded antibiotics, decreasing the dose and administration frequency that are necessary to exert therapeutic efficacy. In particular, liposomes are highlighted as effective nanotechnological tools for the delivery of antibacterial agents [20,28,31,32,54,55]. These lipid-based nanosystems ensure the safety and improve the efficacy of loaded compounds, being able to change the pharmacokinetic and biodistribution profile of incorporated antibiotics [28,56]. Moreover, liposomes are able to preferentially accumulate at infected sites and can also enhance the interaction within the biofilm matrix (reviewed in [18,31,32,57]). In the present work, RFB was incorporated with four different lipid compositions, and the obtained data are described in Table 2.

As depicted in Table 2, all RFB liposomal formulations exhibited high loading capacity (36–43 μg/µmol) and I.E. (43–55%). Previous work with RFB-loaded DMPC:DMPG (8:2) [20] and DMPC:DMPG (7:3) [28] showed similar loading capacity (36 and 47 µg/µmol, respectively) and I.E. (51 and 55%, respectively). Bilayer fluidity is an important factor to consider when designing liposomal formulations. These properties need to ensure the stability of associated compounds and appropriate incorporation parameters. In the present work, in addition to the moderately fluid phospholipids DMPC/DMPG (with a phase transition temperature (Tc) = +23 °C), RFB liposomes using DPPC/DPPG phospholipids that have a Tc = +41 °C were also prepared. Early studies of our research team have demonstrated that the use of DPPC/DPPG lipid mixtures was able to promote high blood residence time of RFB and an effective accumulation at major organs such as liver, spleen, and lungs [28]. Although DMPC/DMPG liposomes may increase the loading capacity and I.E. of RFB, its stability is lower when compared to more rigid bilayers [20,28].

An average mean size below 120 nm was recorded for all RFB liposomes, displaying high homogeneity with a PdI < 0.1 (Table 2 and Appendix A). The inclusion of DSPE-PEG, which confers long blood circulating properties, did not affect the incorporation parameters of liposomes or the mean hydrodynamic size. In terms of surface charge, the presence of this polymer in RFB-LIP1 and RFB-LIP2 resulted in a zeta potential close to neutrality (−5 ± 1 mV), as opposed to the negative charge of RFB-LIP3 (−14 ± 1 mV) and RFB-LIP4 (−15 ± 1 mV). For unloaded liposomes, a low mean size (108–121 nm) and high homogeneity (PdI < 0.1) were also recorded (Table 2 and Appendix A). The zeta potential was equivalent to the RFB-loaded liposomes, with a neutral charge for unloaded-LIP1 and unloaded-LIP2, and a negative zeta potential for unloaded-LIP3 and unloaded-LIP4 (Table 2).

The stability of DSPE-PEG-containing liposomal suspensions, RFB-LIP1 and RFB-LIP2, was assessed after a storage period of 7 days, at 4 °C (Appendix A). Obtained data indicated that the more fluid lipid composition, DMPC:DMPG:DSPE-PEG (RFB-LIP1), displayed a higher % of release of the antibiotic compared to the more rigid one DPPC:DPPG:DSPE-PEG (RFB-LIP2). RFB-LIP1 and RFB-LIP2 showed an RFB release of 35 and 16%, respectively, after 7 days of storage. The higher stability of liposomes composed with lipids of higher Tc (+41 °C) is in accordance with previous reports [28]. Biodistribution studies in Balb/c mice demonstrated that, at 24 h post-intravenous injection, non-metabolized RFB was only detected for liposomes containing rigid phospholipids (DPPC:DPPG) [28].

### 2.4. Susceptibility of Planktonic and Biofilm S. aureus to RFB-Loaded Liposomes

After the successful preparation and characterization of RFB-loaded liposomes, the antibacterial properties of these nanoformulations were assessed against both planktonic and biofilm MRSA-C1 (Figure 5 and Figure 6). As the main goal is to develop a therapeutic strategy against systemic *S*. *aureus* infection, liposomes with prolonged blood circulating times, RFB-LIP1 and RFB-LIP2, were selected for these studies. In parallel, unloaded liposomes with the same lipid compositions were also evaluated at the corresponding lipid concentrations.

As shown in Figure 5 and Figure 6, results demonstrated that both RFB-LIP1 and RFB-LIP2 retained the antibacterial activity of RFB in the free form against planktonic and biofilm bacteria. As detailed in Table 3, MIC values of 0.009, 0.013, and 0.013 μg/mL were obtained for Free RFB, RFB-LIP1, and RFB-LIP2, respectively. Also, MBIC_50_ values of 0.010, 0.008, and 0.008 μg/mL were achieved for Free RFB, RFB-LIP1, and RFB-LIP2, respectively. In turn, unloaded liposomes tested at the corresponding lipid concentrations did not affect bacteria growth. This confirmed that the antibacterial effect was due to RFB action since unloaded liposomes were innocuous. 

Several studies conducted on *S*. *aureus* reinforce the advantages of using antibiotic-loaded liposomes for enhanced antibacterial effect and improved safety, compared to the free drug (reviewed in [30,31,32,33]). Liposomes depending on their physicochemical properties are able to interact with biofilm-organized bacteria since they can penetrate the matrix and release loaded drugs [53,57]. Importantly, the release of associated antibiotics from liposomes is further promoted by MRSA-secreted toxins that upon insertion into the bilayer are able to create pores and facilitate drug leakage [58]. 

The successful antibiofilm activity of liposomes depends on several factors, including the presence of PEG at the liposomal surface [57]. In a study comparing liposomes with and without PEG, researchers concluded that both formulations improved the antibacterial activity of nafcillin towards planktonic and biofilm MSSA, with PEGylated liposomes exerting a more potent effect (2-fold) compared to non-PEGylated ones [59]. Moreover, in biofilms of *Pseudomonas aeruginosa* and *Staphylococcus epidermis*, the inclusion of PEG in the lipid composition also increased [60] or maintained [61] the antibiofilm effectiveness of liposomes compared to those without the polymer. In the present study, RFB-LIP1 and RFB-LIP2, containing DSPE-PEG in the lipid composition, preserved the antibiofilm activity of Free RFB (Table 3).

Recent studies with antibiotic-loaded liposomes confirm the potential of this strategy against *S*. *aureus*. A work from Ferreira et al. (2021) involved the design of RFB liposomes with different lipid compositions, and the assessment of in vitro antibacterial activity against planktonic and biofilm MSSA [20]. Compared to Ferreira et al. (2021) [20], in our study, we focused on the design of RFB-loaded liposomes including DSPE-PEG and tested the antibacterial effectiveness using clinical MRSA strains for in vitro and in vivo assays. 

Researchers have also explored liposomal formulations of different antibiotics. For instance, Rani et al. (2022) have designed liposomes co-loading VCM and daptomycin, with an external surface mimicking the red blood cells membrane [62]. The authors only evaluated the antibacterial activity against planktonic bacteria, while our study also presents biofilm results. In addition, we performed the in vivo therapeutic evaluation in MRSA systemic models, while Rani et al. (2022) conducted biodistribution and safety evaluation in healthy rats [62]. Moreover, *S*. *aureus* strains resistant to VCM and daptomycin are emerging [63], contrary to RFB that is currently applied to treat mycobacteria infections and, in the present work, the main goal was to validate the repurpose of this antibiotic towards *S*. *aureus* infection.

Ashar and colleagues (2023) have developed temperature-sensitive liposomes loading ciprofloxacin [64]. This liposomal formulation, with a mean size of 184 nm, was tested in a rat model of implant-associated MRSA biofilm osteomyelitis. After intravenous administration of liposomes, high-intensity focused ultrasound was used to promote the antibiotic release at the site of bone infection [64]. While Ashar et al. (2023) only assessed biofilm infection targeting, in the present study, smaller liposomes (around 100 nm) loading RFB were designed and these nanoformulations were shown to be effective against both planktonic and biofilm MRSA, without the need to resort to additional external stimulus and/or equipment.

The research team of Makhathini and collaborators (2019) developed VCM-loaded pH-responsive liposomes containing newly synthetized fatty acid-based zwitterionic lipids [65]. The different liposomal formulations were tested in vitro against planktonic MRSA and the proof-of-concept was conducted in a murine model of skin infection with the tested formulations locally administered into the infection site [65]. In the current work, systemic models of MRSA infection were established, representing a more advanced infection status. Furthermore, for the preparation of liposomal formulations, we used phospholipids that are currently present in several approved nanoformulations [66], while Makhathini et al. (2019) synthetized new lipids to design pH-sensitive liposomes [65]. Overall, these recent works within liposomes as antibiotic nanocarriers highlight the investment in this lipid-based nanosystem. Liposomes prove to be a versatile and advantageous tool for the development of more effective and safe therapeutic approaches against *S*. *aureus*.

### 2.5. Interaction of Unloaded and RFB-Loaded Liposomes with S. aureus Biofilm by Microscopy

As aforementioned, to overcome current challenges in antibiofilm therapy, the association of antibiotics within liposomes constitutes an advantageous strategy to facilitate drug penetration into *S*. *aureus* biofilm, increasing the local concentration of antibiotic [20,53,67]. Several reports have proven the importance of liposomes as antibiotic delivery systems, allowing to overcome bacteria resistance to drugs in the free form (reviewed in [31,32,57]). 

Here, to deeply understand the interaction of liposomes with the biofilm structure, we resorted to confocal laser scanning microscopy (CLSM) to obtain high-resolution images of biofilms at different depths and planes. For this analysis, both unloaded and RFB-loaded liposomes labeled with rhodamine were incubated for 24 h. As depicted in Figure 7 and Appendix A, the presence of RFB did not hinder the interaction of liposomes with the biofilm matrix. Furthermore, while unloaded liposomes did not affect biofilm growth, the presence of RFB led to an evident reduction in biofilm thickness. These data were in accordance with the MTT assay, where RFB-loaded liposomes greatly decreased bacteria viability in a concentration-dependent manner, and unloaded liposomes were innocuous (Figure 6). Previously, our research group has reported the successful interaction and internalization of liposomes within *S*. *aureus* biofilms [20]. In particular, the positively charged liposomes DMPC:stearylamine (+13 mV), with a mean size 130 nm, showed high biofilm interaction [20]. This has also been observed by other researchers evaluating liposomes composed of distearoyl phosphatidyl choline (DSPC) and dioleyl trimethylammonium propane (DOTAP) [68]. These small mean size liposomes (<130 nm) with a very high positive charge (+59 mV) have also exhibited elevated levels of penetration in MSSA biofilms [68].

After assessing the behavior of liposomal formulations in MRSA-C1 biofilms, the next step was to investigate the kinetics of this interaction. This analysis was accomplished with rhodamine-labeled unloaded-LIP2 that was incubated with the biofilm at a lipid concentration of 1.5 μmol/mL. The distribution of liposomes was monitored during 120 min by CSLM (Figure 8a) and the fluorescence intensity signal was normalized in relation to time = 0 min (Figure 8b). The results showed that liposomes internalization within the biofilm structure was a rapid event, with a sharp increase in fluorescence intensity in the first 30 min, followed by a steady phase up to 120 min. The high antibiofilm efficacy of tested nanoformulation was supported by confocal microscopy, showing that liposomes successfully reached the inner layers of biofilm, releasing RFB within the vicinity of target bacteria (Figure 8c). 

### 2.6. Therapeutic Evaluation of RFB Formulations against Systemic MRSA Infection

Considering the promising in vitro results of RFB formulations, their therapeutic potential was evaluated in murine models of systemic MRSA-C1 infection. In the first study (designated as In Vivo Assay 1), mice were infected with 1.4 × 10^9^ CFU/mouse and only Free RFB and RFB-LIP2 were evaluated (Figure 9). In a second infection model (In Vivo Assay 2), induction was performed with 3.4 × 10^8^ CFU/mouse and evaluated formulations were Free RFB, RFB-LIP1, and RFB-LIP2, as well as the positive control, VCM (Appendix A). The rationale for selecting these two lipid compositions was the inclusion of DSPE-PEG. On the one hand, DSPE-PEG promotes long blood circulating properties, maximizing the extravasation at infected sites. On the other hand, these types of lipid compositions have been successfully employed for delivering loaded compounds to infected sites, namely mycobacterial, bacterial, and parasitic infections [29,55,56]. These previous results demonstrated the therapeutic effectiveness of developed liposomal formulations and their favorable biodistribution profile in major organs, such as the spleen, liver, and lungs, which are also some of the most affected ones in MRSA systemic infection. Altogether, these factors were taken into consideration for the selection of the two lipid compositions containing DSPE-PEG, RFB-LIP1, and RFB-LIP2, for the in vivo proof-of-concept.

Groups treated with RFB formulations at a dose of 20 mg/kg exhibited 100% survival, while the Control group reached 50% survival 7 days after infection induction (Figure 9a). This was correlated with a recovery in body weight by animals receiving RFB treatment that, at day 7 post-infection induction, displayed a significantly higher body weight than the Control group (Figure 9b). RFB formulations also led to a significant reduction in bacterial burden (Figure 9c) and growth index values (Figure 9d), compared to Control group mice. Importantly, the antibacterial effect of RFB was greatly enhanced when incorporated in liposomes. RFB-LIP2 treatment significantly decreased bacterial burden and growth index in all four major organs (liver, spleen, kidneys, and lung), while for Free RFB, such an effect was only observed in the spleen and kidneys.

Following the first in vivo assay (Figure 9), in the In Vivo Assay 2, RFB formulations also demonstrated a potent therapeutic effect against systemic MRSA-C1 infection, as shown in Appendix A. Mice treated with RFB in the free and liposomal forms (RFB-LIP2) displayed 100% survival, compared to 80% for VCM and 60% for Control (Figure 9a). The higher survival rate observed for the Control group, when compared to the In Vivo Assay 1 (60% vs. 50%), might be due to the lower infection dose used (3.4 × 10^8^ CFU/mouse vs. 1.4 × 10^9^ CFU/mouse, respectively). In addition, the 100% survival rate observed for animals treated with RFB formulations was correlated with a recovery in body weight, which was not seen for positive and negative controls, as depicted in Appendix A. Furthermore, in terms of bacterial burden (Appendix A) and growth index (Appendix A), a significant reduction was attained in major organs of mice administered with RFB formulations. In the spleen, the Control group displayed a growth index of 0.07 ± 0.90, while Free RFB, RFB-LIP1, RFB-LIP2, and the positive control, VCM, presented values of −2.38 ± 0.37, −3.30 ± 0.19, −2.92 ± 0.31, and −2.29 ± 0.18, respectively. In the kidneys, the growth index of Control mice was −0.23 ± 0.45, and for the groups Free RFB, RFB-LIP1, RFB-LIP2, and VCM, values were −3.11 ± 0.62, −2.79 ± 0.35, −3.31 ± 0.29, and −0.86 ± 0.31, respectively. In the liver, only RFB-LIP2 significantly reduced the bacterial burden and growth index. It should be mentioned that the antibacterial effect of RFB formulations was achieved at a 2-fold lower dose than the positive control, VCM, further evidencing their antibacterial potential. Furthermore, the superior therapeutic effectiveness of RFB-LIP2 was evidenced in the model that was induced with a 4-fold higher MRSA inoculum, achieving a significant antibacterial effect in all tested organs, compared to Control and Free RFB (Figure 9). These results are of extreme importance given that infection sites may be exposed to subtherapeutic concentrations of antibiotics when in the free form. This may contribute to the emergence of resistance and cause severe systemic toxicity [18,31]. Despite this situation being aggravated by a higher inoculum, liposomal RFB was able to overcome these challenges and exert a significant therapeutic effect, with no adverse events.

In the In Vivo Assay 2, the histopathological analysis of liver, spleen, kidneys, and lung was performed using H&E staining (Figure 10). Systemic *S*. *aureus* infection can form abscesses and persist in different organs. The kidneys are frequently the most affected ones, displaying high bacterial burdens [69]. In the present study, the main lesions were identified in the liver and kidneys as foci of neutrophil infiltration with abscess formation. In the liver, the abscesses were multifocal and small. In the kidneys, the inflammatory infiltrates were mainly located in the medulla, and the Control and VCM groups exhibited the highest scores for inflammation/necrosis (Appendix A). In the spleen, the presence of germinal center apoptotic bodies was indicative of reactivity (Appendix A). 

Tissue index was evaluated as a safety parameter in both models of systemic *S*. *aureus* infection (Appendix A). Organ weight changes are frequently assessed as an indicator of toxicity. On the one hand, increased tissue index values may be associated with hypertrophy, congestion, or edema. On the other hand, lower values are indicative of organ atrophy or degeneration [70,71,72]. In the present study, among the different experimental groups, no major changes in the analyzed organs were detected compared to naïve mice, thus demonstrating the safety of tested formulations.

As aforementioned, the use of liposomes aims to improve the efficacy and/or the safety of associated drugs. Antibiotics in the free form display low bioavailability and concentration at infected sites, a consequence of their unfavorable biodistribution profile [18,31]. Only a fraction of the originally administered antibiotic dose reaches the target, and this may influence the emergence of resistance [18,31]. The low concentration of antibiotic at infected sites requires frequent administrations of high doses in order to exert a therapeutic effect, resulting in the origin of toxic side effects [31]. To overcome these hurdles, liposomes offer many advantages as antibiotic nanocarriers. These lipid-based nanosystems may reduce dose and frequency of administrations, prevent enzymatic degradation, avoid drug efflux mechanisms, and overcome outer membranes [18,31,73,74]. In the case of RFB, previous studies in a murine model of disseminated *Mycobacterium tuberculosis* infection demonstrated a significantly superior antimycobacterial effect of RFB-loaded liposomes in the spleen and liver, compared to free antibiotic [28]. Moreover, compared with Free RFB, lung inflammation was reduced in mice treated with liposomal RFB, which indicated a better safety profile [28].

Importantly, liposomes are advantageous for the eradication of biofilm-organized bacteria, such as those formed in orthopedic implants and medical devices, promoting the penetration within the biofilm and release of the antibacterial agent [18,31,73]. Moreover, different lipid compositions can be designed to maximize the preferential accumulation of liposomal formulations at diseased sites, including the functionalization of surface with specific moieties for a targeted approach [75,76]. Notably, in the present work, only one dose and one treatment schedule were evaluated. Different therapeutic regimens could be further assessed to optimize the therapeutic index of RFB formulations.

## 3. Materials and Methods

### 3.1. Reagents

Phosphate-buffered saline (PBS) and VCM were purchased from Sigma-Aldrich (St. Louis, MO, USA) and RFB from Pharmacy Biotech AB (Uppsala, Sweden). The pure phospholipids, dimyristoyl phosphatidyl choline (DMPC), dimyristoyl phosphatidyl glycerol (DMPG), dipalmitoyl phosphatidyl choline (DPPC), dipalmitoyl phosphatidyl glycerol (DPPG), and distearoyl phosphatidyl ethanolamine covalently linked to poly(ethylene glycol)2000 (DSPE-PEG) were purchased from Lipoid (Ludwigshafen, Germany). Rhodamine covalently linked to phosphatidylethanolamine (Rho-PE) were purchased from Avanti Polar Lipids (Alabaster, AL, USA). Thiazolyl blue tetrazolium bromide (MTT), crystal violet (CV), and glycerol were obtained from Panreac AppliChem, ITW Reagents (Darmstadt, Germany). Culture media Mueller–Hinton Agar (MHA) and Mueller–Hinton Broth (MHB) were obtained from Oxoid, Ltd. (Basingstoke, UK) and Tryptic Soy Broth (TSB) from Biokar (Pantin, France). The fluorescent stain SYTO 9 was obtained from Molecular Probes (Eugene, OR, USA). D(+)-glucose monohydrate was acquired from Merck KGaA (Darmstadt, Germany). Dimethyl sulfoxide (DMSO) and ethanol absolute anhydrous were obtained from Carlo Erba Reagents S.A.S. (Val de Reuil, France). All other reagents were of analytical grade.

### 3.2. Animals

Male Balb/c mice (6–8 weeks old) were obtained from Charles River Laboratories (Barcelona, Spain). Animals were kept under standard hygiene conditions, in ventilated cages on a 12 h light/12 h dark cycle, at 20–24 °C and 50–65% humidity. Mice had free access to sterilized diet and acidified water. All animal experiments were conducted according to the Animal Welfare Organ of the Faculty of Pharmacy, Universidade de Lisboa, approved by the competent national authority *Direção-Geral de Alimentação e Veterinária* (DGAV) and in accordance with the EU Directive (2010/63/UE) and Portuguese laws (DR 113/2013, 2880/2015, 260/2016 and 1/2019) for the use and care of animals in research.

### 3.3. Preparation of RFB-Loaded Liposomes

A remote loading technique, based on an ammonium sulfate gradient, was used to encapsulate RFB in pre-formed empty liposomes, as previously described [28]. The tested lipid compositions were DMPC:DMPG:DSPE-PEG (65:30:5; LIP1), DPPC:DPPG:DSPE-PEG (65:30:5; LIP2), DMPC:DMPG (80:20; LIP3), and DPPC:DPPG (80:20; LIP4). Briefly, in a round-bottomed flask, selected phospholipids were dissolved in chloroform and dried in a rotary evaporator (Buchi R-200, Flawil, Switzerland) to form a thin lipid film. Afterwards, the lipid film was dispersed with deionized water for a final lipid concentration of 30 μmol/mL. The obtained lipid suspensions were frozen at −70 °C and lyophilized (freeze-dryer, Edwards, CO, USA) overnight. The lyophilized product was rehydrated with a buffer solution containing 120 mM of ammonium sulphate (pH 5), within a temperature set above the phase transition temperature (T_c_) of the selected phospholipids. Then, using an extruder device (Lipex: Biomembranes Inc., Vancouver, Canada), the dispersions were sequentially filtered through polycarbonate membranes, under nitrogen pressure (10–500 lb/in2), to achieve an average vesicle size around 100 nm. An ammonium sulfate gradient was established by replacing the extraliposomal medium with HEPES buffer pH 6.9 (10 mM HEPES, 140 mM NaCl), using a desalting column (Econo-Pac^®^ 10 DG; Bio-Rad Laboratories, Hercules, CA, USA). An RFB solution at 0.5 mg/mL was prepared in the same buffer and, subsequently, incubated at 100 nmol/μmol of lipid with unloaded liposomes for 1 h, under stirring, at a temperature above the T_c_ of the phospholipids mixture. Non-encapsulated RFB was removed by ultracentrifugation at 250,000× *g*, for 120 min, at 15 °C in a Beckman LM-80 ultracentrifuge (Beckman Instruments, Inc., Fullerton, CA, USA). The obtained pellets were suspended in HEPES buffer pH 6.9. Unloaded liposomes were prepared following the same methodology. Fluorescent liposomes were obtained by including in the lipid composition Rho-PE at 0.1 mol% [77].

### 3.4. Characterization of RFB-Loaded Liposomes

Liposomes were characterized in terms of RFB and lipid contents, mean size, and zeta potential. For RFB quantification by spectrophotometry, liposomes were disrupted with ethanol and absorbance was read at 500 nm [28]. Phospholipid content was determined following a colorimetric technique defined by Rouser [78]. Loading capacity was defined as the final RFB to lipid ratio (RFB/Lip)f and the incorporation efficiency (I.E.), in percentage, was determined according to Equation (1):(1)$$I.E.\left(\%\right)=\frac{\left(\frac{\mathrm{RFB}}{\mathrm{Lip}}\right)f}{\left(\frac{\mathrm{RFB}}{\mathrm{Lip}}\right)i}\;\times \;100$$

The mean size of liposomes and respective polydispersity index (PdI) were determined by dynamic light scattering in a Zetasizer Nano S (Malvern Instruments, Malvern, UK). The zeta potential was determined by laser doppler spectroscopy in a Malvern Zetasizer Nano Z (Malvern Instruments, Malvern, UK).

### 3.5. S. aureus Strains and Culture Conditions

Clinical MRSA isolates, MRSA-C1 and MRSA-C2, were kindly provided by Centro Hospitalar Universitário de Lisboa Central (CHULC, EPE), under a collaborative project. From 24 h bacterial cultures in MHA, stocks were prepared in MHB with 20% glycerol and stored at −80 °C. From these frozen stocks and for each assay, fresh cultures were grown 24 h in MHA, at 37 °C. 

### 3.6. Susceptibility of Planktonic S. aureus to Antibiotics

The antibacterial activity of RFB formulations and VCM was evaluated by the broth microdilution assay, in accordance with the guidelines of the Clinical and Laboratory Standards Institute [79], followed by turbidity assessment [20]. RFB was quantified by spectrophotometry at 500 nm [28] and VCM concentration was determined using the Lowry and Folin method [80,81]. RFB and VCM formulations diluted in MHB were tested at concentrations ranging from 0.0016 to 0.414 μg/mL and 0.014 to 3.750 μg/mL, respectively. To prepare the inoculum, bacteria from 24 h cultures were diluted in MHB and, by measuring the optical density (OD) at 600 nm (Shimadzu UV 160A, Shimadzu Corporation, Kyoto, Japan), turbidity was adjusted to 0.5 in a McFarland scale, corresponding to, approximately, 10^8^ colony-forming units per mL (CFU/mL). Bacterial suspensions were placed in non-adherent 96-well U-bottom polystyrene plates for a final concentration of 5 × 10^5^ CFU/mL (MRSA-C1 and MRSA-C2). Tested formulations were added to the respective wells and incubated at 37 °C, for 24 h, under static conditions. Negative controls were bacteria in MHB without antibiotics and sterile controls were MHB alone. Empty liposomes were also assayed using the same lipid concentrations as the ones tested for the respective RFB-loaded liposomes. Minimum inhibitory concentration (MIC) was determined spectrophotometrically at 570 nm in a microplate reader (iMark^TM^, Bio-Rad laboratories, Inc., Hercules, CA, USA), being defined as the lowest antibiotic concentration that inhibited visible bacterial growth. For MRSA-C1, the obtained MIC values of Free RFB and VCM were further confirmed by CFU counting and MTT assay (visualization). For CFU counting, after incubation period, 50 μL of wells (in triplicate) were collected, serially diluted in sterile PBS, seeded in MHA plates, and individual colonies were counted after 24 h at 37 °C. In MTT assay, after the incubation period, plates were centrifuged at 800× *g*, for 10 min, at room temperature. Medium was carefully removed and 200 μL/well of MTT solution at 125 μg/mL in PBS was added, followed by incubation at 37 °C until color formation (5–15 min). The MIC value was defined as the minimum concentration of antibiotic where no coloration was observed, compared to the negative control.

### 3.7. Susceptibility of S. aureus Biofilm to Antibiotics

Cultures of 24 h of each strain, MRSA-C1 and MRSA-C2, were diluted in TSB 0.25% and placed in 96-well F-bottom polystyrene plates at a final concentration of 1 × 10^6^ CFU/mL (200 μL/well) [20]. Biofilms were grown at 37 °C, for 24 h, under static conditions. Tested antibiotic formulations were added and incubated for 24 h, at 37 °C, with RFB and VCM at concentrations ranging from 0.0008 to 0.828 μg/mL and 6.25 to 800 μg/mL, respectively. Negative controls were bacteria in TSB 0.25% without antibiotics and sterile controls were TSB 0.25% alone. Unloaded liposomes were also tested using the lipid concentrations corresponding to the respective RFB-loaded liposomes. 

The in vitro activity of tested antibiotic formulations was assessed through the MTT assay, as described in literature [82], with some modifications. After rinsing the attached bacteria twice with PBS (200 μL/well), MTT solution at 125 μg/mL in PBS was added (200 μL/well) and incubated at 37 °C, for 1 h. Afterwards, the MTT solution was discarded and 200 μL of DMSO were added to each well to solubilize the purple formazan crystals. The OD was measured at 570 nm in a microplate reader (iMark^TM^, Bio-Rad laboratories, Inc., Hercules, CA, USA). Bacteria viability, in percentage, was calculated according to Equation (2), where OD_t_ refers to the OD values of bacteria incubated with the tested formulations, and OD_ctrl_ are the OD values of negative control that correspond to 100% viability.
(2)$$\mathrm{Bacteria}\;\mathrm{viability}\;(\%)=\left(\frac{{\mathrm{OD}}_{t}}{{\mathrm{OD}}_{\mathrm{ctrl}}}\right)\times 100\;$$

Minimum biofilm inhibitory concentration (MBIC) was defined as the lowest antibiotic concentration able to inhibit 50% of bacterial growth, relative to negative controls (MBIC_50_). The determination of MBIC_50_ was performed by sigmoidal fitting analysis using GraphPad Prism version 8.0.1 for Windows (GraphPad Software, San Diego, CA, USA).

Quantification of biofilm biomass was performed through CV staining method [20,83]. Briefly, medium was discarded and attached bacteria were washed twice with PBS (200 μL/well), followed by drying at room temperature for 15 min. Then, biofilms were incubated with 200 μL of a CV solution at 0.0125% (*w*/*v* in water) at room temperature, during 10 min, followed by two washing steps with PBS. After drying at 37 °C for 10 min, stained biofilms were dissolved with 200 μL ethanol and absorbance was measured at 570 nm using a microplate reader (iMark^TM^, Bio-Rad laboratories, Inc., Hercules, CA, USA). Negative control was the biofilm in the presence of TSB 0.25% only, representing 100% growth. Biofilm biomass (%) was calculated using Equation (3), where OD_t_ corresponds to the biofilms incubated with the tested formulations and OD_ctrl_ refers to the negative control (100% biofilm biomass).
(3)$$\mathrm{Biofilm}\;\mathrm{Biomass}\;(\%)=\left(\frac{{\mathrm{OD}}_{t}}{{\mathrm{OD}}_{\mathrm{ctrl}}}\right)\times 100\;$$

### 3.8. Interaction of RFB Liposomes with S. aureus Biofilm by Microscopy

Confocal scanning laser microscopy (CSLM) was used to evaluate the interaction of developed liposomal formulations with 24 h-old MRSA-C1 biofilms. For this assay, liposomes fluorescently labeled with Rho-PE (unloaded and RFB-loaded) were used. Biofilms were established in 8-well chambered coverslips (Ibidi GmbH, Munich, Germany), with 200 μL/well of MRSA-C1 bacteria inoculum at 1 × 10^6^ CFU/mL in TSB supplemented with 0.25% of glucose (TSB 0.25%), and incubated at 37 °C, for 24 h [84]. For these assays, liposomes at a lipid concentration of 1.5 µmol/mL in TSB 0.25% were used. Control biofilm corresponded to bacteria in TSB 0.25% medium.

In the first assay, RFB-LIP1, RFB-LIP2, RFB-LIP3, RFB-LIP4, as well as unloaded liposomes were incubated for 24 h with MRSA-C1 biofilms. Afterwards, biofilms were washed with PBS and stained with SYTO 9 (3 µM), for 30 min in the dark, at room temperature. In the kinetics study, mature biofilms were stained with SYTO 9 (3 µM), incubated with unloaded-LIP2 and monitored over a period of 120 min, with 10 min intervals. Biofilms were visualized using a Leica TCS SP5 inverted microscope (Leica Mycrosystems CMS GmbH, Mannheim, Germany) equipped with a continuous Ar ion laser (Multi-line LASOS^®^ LGK 7872 ML05). Image acquisition was performed at 512 by 512 pixels, with a scan rate of 100 Hz per frame. A 63 × 1.2 N.A. water immersion objective was used (HCX PL APO CS 63.0 × 1.20 WATER UV). SYTO 9 images were recorded with a 488 nm excitation line and emission was recorded at 501–570 nm. Rho-PE images were collected with a 514 nm excitation line and emission was recorded at 610–760 nm. For each condition, planes xzy and xyz were analyzed. 

### 3.9. Therapeutic Evaluation of RFB Formulations in Murine Models of Systemic MRSA Infection

The antibacterial effect of RFB formulations was assessed in systemic murine models of MRSA-C1 infection using male Balb/c mice (6–8 weeks old), obtained from Charles River Laboratories (Barcelona, Spain). Two independent infection models were tested: 1.4 × 10^9^ CFU/mouse, designated as In Vivo Assay 1, and 3.4 × 10^8^ CFU/mouse, corresponding to In Vivo Assay 2. Bacteria inoculum was given intravenously in the lateral tail vein. In the In Vivo Assay 1 (1.4 × 10^9^ CFU/mouse), only Free RFB and RFB-LIP2 were assessed. In the In Vivo Assay 2 (3.4 × 10^8^ CFU/mouse), Free RFB, RFB-LIP1, RFB-LIP2, and VCM were evaluated. RFB formulations were administered at 20 mg/kg and VCM at a dose of 40 mg/kg. Control mice group received buffer by intravenous route.

In both models, three days after infection induction, 3 to 5 animals were sacrificed and organs of interest (liver, spleen, kidneys, and lungs) were aseptically collected, homogenized, and serially diluted in PBS for CFU counting in MHA plates. Colonies were counted after 24 h of incubation at 37 °C. Growth index was determined as the difference between the log_10_CFU at the end of the protocol and the log_10_CFU at the beginning of treatment. Tissue index was calculated using Equation (4):(4)$$\mathrm{Tissue}\;\mathrm{index}=\sqrt{\frac{\mathrm{organ}\;\mathrm{weight}}{\mathrm{animal}\;\mathrm{weight}}}\times 100\;$$

Simultaneously, the treatment schedule was initiated. Groups received IV administrations of tested formulations, once per day, for three consecutive days. Two days after the last treatment, mice were sacrificed and liver, spleen, kidneys, and lung were collected and processed for CFU counting as abovementioned. For histology, organs were fixed in 10% neutral buffered formalin, and sections were stained and examined by the H&E method. Whole-slide images were obtained using the NanoZoomer-SQ Digital slide scanner (Hamamatsu Photonics, Shizuoka, Japan). Tissue indexes were calculated using Equation (4). The antibacterial effectiveness of the formulations was determined in terms of average body weight, percentage of survival, bacterial burden, and growth index.

### 3.10. Statistical Analysis

Results are expressed as mean ± standard deviation (SD) or standard error mean (SEM). Statistical analysis was performed with one-way or two-way ANOVA followed by Dunnet’s post hoc test using GraphPad Prism version 8.0.1 for Windows (GraphPad Software, San Diego, CA, USA). *p* < 0.05 was considered statistically significant.

## 4. Conclusions

The present results are very encouraging, emphasizing the application of liposomes for the delivery of RFB against *S*. *aureus* infections, especially for the more resistant biofilm-organized bacteria. We successfully designed four liposomal formulations using phospholipids with distinct T_c_ (DMPC/DMPG = +24 °C/+23 °C and DPPC/DPPG = +41 °C) with high RFB loadings. DSPE-PEG was included in the lipid composition to achieve prolonged blood circulation times. In vitro studies demonstrated that RFB retained its antibacterial effect when nanoformulated. Importantly, RFB nanoformulations were more effective than the gold-standard antibiotic, VCM. Moreover, the antibacterial activity of RFB liposomes against planktonic and biofilm-organized bacteria was confirmed to be solely due to the antibiotic presence since unloaded liposomes did not exert any effect. 

The antibiofilm activity of RFB liposomes was correlated with the successful interaction of nanoformulations with MRSA-C1 biofilm as demonstrated by confocal microscopy studies. Furthermore, the presence of either DSPE-PEG and/or RFB did not hinder this interaction. Following these promising in vitro data, the proof-of-concept was performed in murine models of systemic MRSA-C1 infection. A significant reduction in bacterial burden and improvement in survival rates was achieved for mice treated with RFB formulations using a 2-fold lower dose than VCM, thus evidencing the potential of this therapeutic strategy. Continuous efforts should be made to provide new solutions against MRSA, particularly biofilm-organized bacteria, and RFB formulations emerge as an effective and safe option.

## Figures and Tables

**Figure 1 pharmaceuticals-17-00470-f001:**
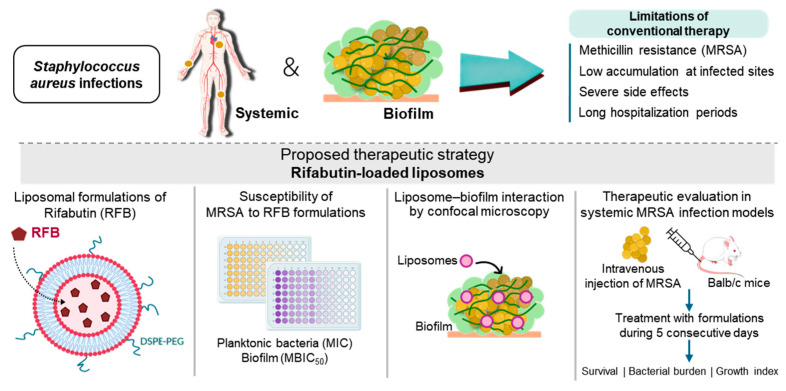
Schematic representation of the experimental design for the development of RFB—loaded liposomes as a therapeutic strategy against MRSA infections.

**Figure 2 pharmaceuticals-17-00470-f002:**
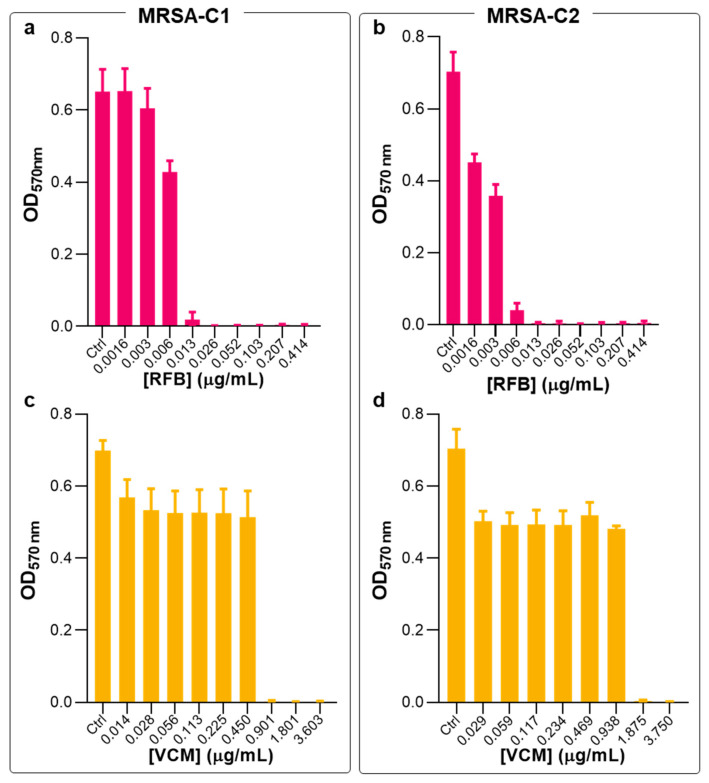
Susceptibility of planktonic MRSA-C1 and MRSA-C2 to RFB and VCM in the free form. The broth microdilution method, followed by turbidity measurement, was performed 24 h after incubation with antibiotics. Bacteria in MHB corresponds to negative control (Ctrl). RFB (**a**,**b**) and VCM (**c**,**d**). RFB and VCM concentrations ranged from 0.0016 to 0.414 μg/mL and 0.014 to 3.750 μg/mL, respectively. Data are expressed as mean ± SD (n = 3–4).

**Figure 3 pharmaceuticals-17-00470-f003:**
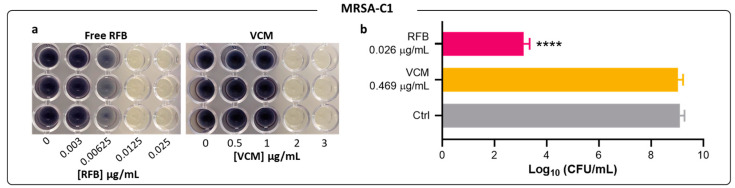
Susceptibility of planktonic MRSA-C1 to Free RFB and VCM. (**a**) Representative images of MTT assay 24 h after incubation with the antibiotics in the free form. (**b**) CFU countings of selected antibiotic concentrations were performed to determine viable bacteria recovered after the same incubation period (24 h). Concentrations of RFB and VCM ranged from 0.0008 to 0.026 μg/mL and 0.117 to 3.75 μg/mL, respectively. Bacteria in MHB corresponds to negative control. One-way ANOVA with Dunnet’s test. **** *p* < 0.001 vs. Ctrl. Results are expressed as mean ± SD (n = 3).

**Figure 4 pharmaceuticals-17-00470-f004:**
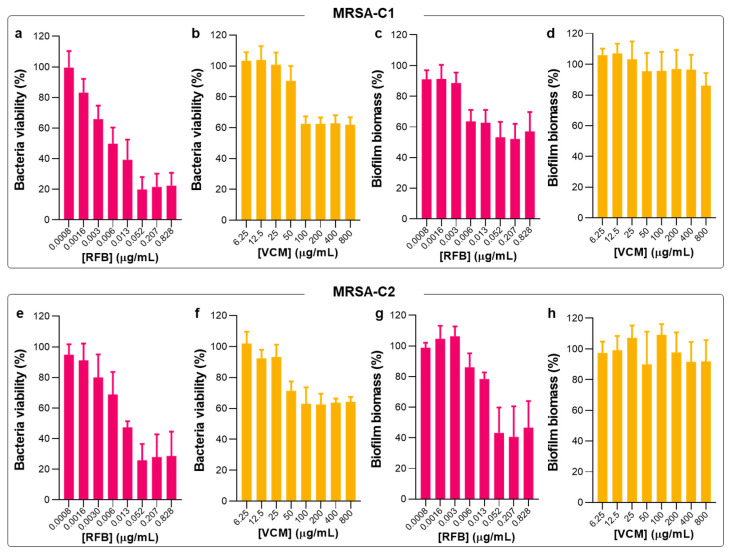
Susceptibility of biofilm MRSA-C1 and MRSA-C2 to Free RFB and VCM. (**a**,**b**,**e**,**f**) Bacteria viability (%) was assessed through the MTT assay and MBIC_50_ was defined as the antibiotic concentration that inhibits 50% of bacterial growth related to negative control. (**c**,**d**,**g**,**h**) Biofilm biomass (%) was assessed through the CV assay. Mature biofilms were incubated with RFB and VCM at concentrations ranging from 0.0008 to 0.828 μg/mL and 6.25 to 800 μg/mL, respectively. Bacteria in TSB 0.25% corresponds to negative controls (100% growth). Results are expressed as mean ± SD (n = 3).

**Figure 5 pharmaceuticals-17-00470-f005:**
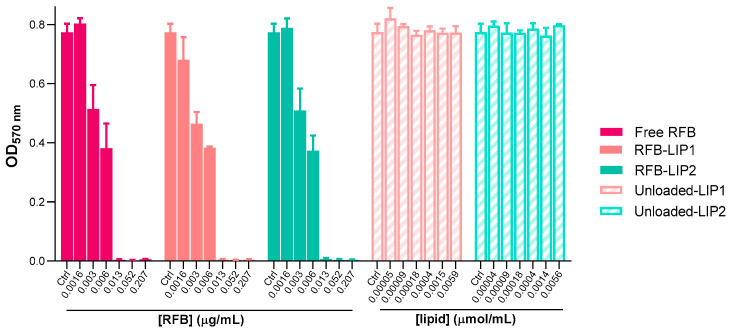
Susceptibility of planktonic MRSA-C1 to RFB formulations, Free RFB, RFB-LIP1, and RFB-LIP2. RFB concentrations ranged from 0.0016 to 0.207 μg/mL. Unloaded liposomes were tested at the same lipid concentrations as corresponding RFB-loaded liposomes. Bacteria in MHB corresponds to negative controls (100% growth). Results are expressed as mean ± SD (n = 3). The susceptibility to tested formulations was assessed by turbidimetry. Lipid compositions: DMPC:DMPG:DSPE-PEG (RFB-LIP1 and Unloaded-LIP1); DPPC:DPPG:DSPE-PEG (RFB-LIP2 and Unloaded-LIP2).

**Figure 6 pharmaceuticals-17-00470-f006:**
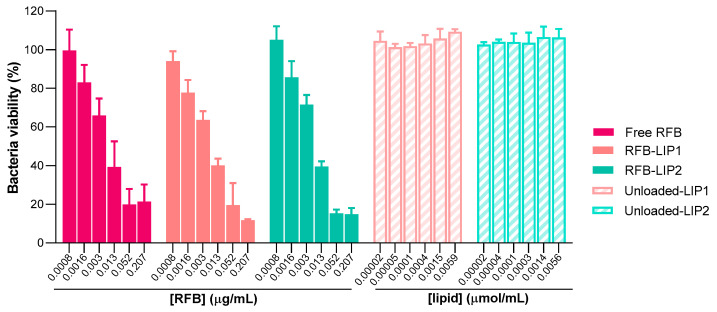
Susceptibility of biofilm MRSA-C1 to RFB formulations, Free RFB, RFB-LIP1, and RFB-LIP2. RFB concentrations ranged from 0.0008 to 0.207 μg/mL. Unloaded liposomes were tested at the same lipid concentrations as corresponding RFB-loaded liposomes. Bacteria in TSB 0.25% corresponds to negative controls (100% growth). Results are expressed as mean ± SD (n = 3). The susceptibility to tested formulations was assessed by MTT assay. Lipid compositions: DMPC:DMPG:DSPE-PEG (RFB-LIP1 and Unloaded-LIP1); DPPC:DPPG:DSPE-PEG (RFB-LIP2 and Unloaded-LIP2).

**Figure 7 pharmaceuticals-17-00470-f007:**
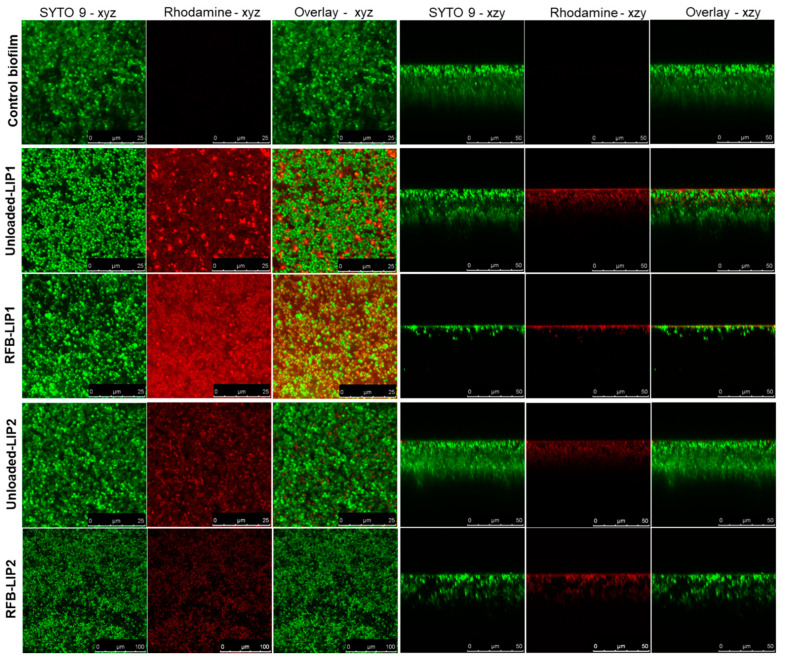
Representative CSLM images of mature MRSA-C1 biofilms after 24 h incubation with rhodamine-labeled LIP1 and LIP2 liposomes, unloaded and RFB-loaded, at a lipid concentration of 1.5 µmol/mL. Biofilms were stained with the green dye SYTO 9 at 3 µM. Untreated biofilm was used as a negative control (Control biofilm). Images in the left panels correspond to *x–y* plane images taken at the inner layer of MRSA-stained biofilms, and images in the right panels correspond to *x–z* orthogonal plane images. The overlay of the green and red channels from each plane image is presented as Overlay—*x*–*y* and Overlay—*x*–*z*. Lipid compositions: DMPC:DMPG:DSPE-PEG (RFB-LIP1 and Unloaded-LIP1); DPPC:DPPG:DSPE-PEG (RFB-LIP2 and Unloaded-LIP2).

**Figure 8 pharmaceuticals-17-00470-f008:**
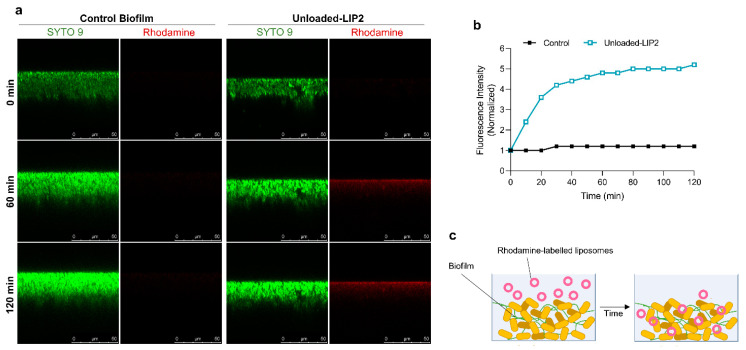
Interaction of rhodamine-labeled unloaded-LIP2 with MRSA-C1 biofilm. Liposomes were incubated at a lipid concentration of 1.5 μmol/mL. Untreated 24 h-old biofilm was used as negative control (Control Biofilm). (**a**) Representative CSLM xzy orthogonal views of MRSA biofilms at time points 0, 60, and 120 min. Biofilms were stained with the green dye SYTO 9 at 3 µM. xzy sections were taken every 10 min for a total of 120 min; (**b**) relative changes in the average of rhodamine fluorescence intensity over time; (**c**) schematic representation of the interaction of rhodamine-labeled liposomes with *S*. *aureus* biofilm. LIP2: DPPC:DPPG:DSPE-PEG.

**Figure 9 pharmaceuticals-17-00470-f009:**
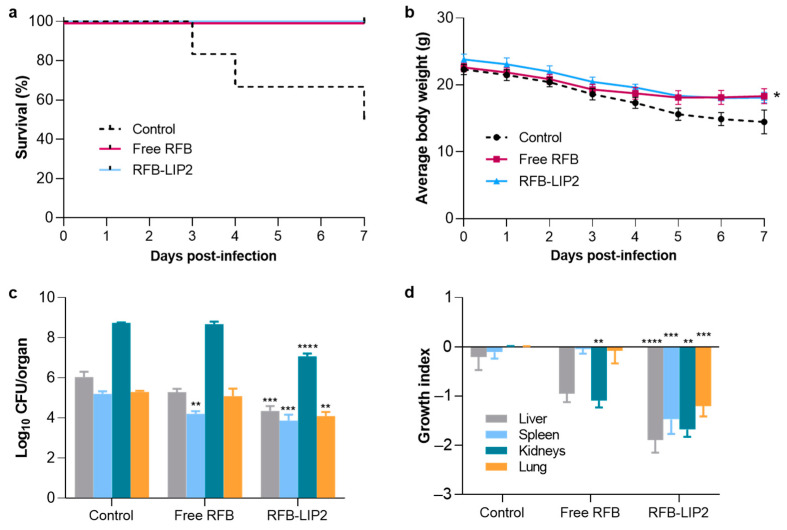
In Vivo Assay 1—Preliminary evaluation of RFB formulations in a murine model of systemic MRSA-C1 infection. Infection was induced intravenously in male Balb/c mice with a MRSA-C1 inoculum at 1.4 × 10^9^ CFU/mouse. Three days after infection induction, mice received IV administrations of RFB formulations (Free RFB and RFB-LIP2) at dose of 20 mg/kg. Control group received buffer by intravenous route. (**a**) Percentage of survival (Kaplan-Meier analysis), (**b**) average body weight, (**c**) bacterial burden in major organs at the end of treatment protocol, and (**d**) growth index. RFB-LIP2: DPPC:DPPG:DSPE-PEG. Results are expressed as mean ± SEM (n = 4–5). Two-way ANOVA with Dunnett’s test. * *p* < 0.05, ** *p* < 0.01, *** *p* < 0.001, **** *p* < 0.0001 vs. Control group.

**Figure 10 pharmaceuticals-17-00470-f010:**
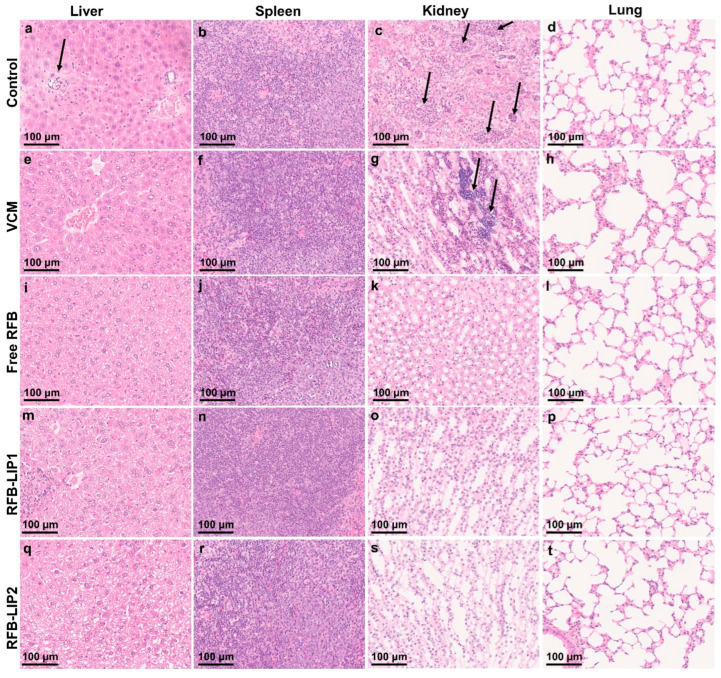
In Vivo Assay 2—Representative images of histological analysis of major organs collected from mice induced with systemic MRSA-C1 infection (3.4 × 10^8^ CFU/mouse): buffer-treated (Control), treated with VCM at a dose of 40 mg/kg of body weight, and with RFB formulations (Free RFB, RFB-LIP1, and RFB-LIP2) at a dose of 20 mg/kg of body weight. Black arrows indicate affected regions. (**a**) Small single abscess; (**c**) extended coalescing foci of abscesses in the medulla with pyelonephritis; (**g**) large abscess in the medulla with presence of bacteria; (**b**,**f**,**j**,**n**,**r**) white pulp germinal center apoptosis minimal; (**s**) minimal inflammatory infiltrates in the interstitium; (**d,h,k,l,o,p,t**) within normal limits; (**e,i**) within normal limits; (**m,q**) multifocal small foci of abscesses. LIP1: DMPC:DMPG:DSPE-PEG; LIP2: DPPC:DPPG:DSPE-PEG. Scale bar = 100 μm.

**Table 1 pharmaceuticals-17-00470-t001:** Susceptibility of planktonic and biofilm *S*. *aureus* strains to RFB and VCM in the free form.

	MIC (μg/mL)		MBIC_50_ (μg/mL)	
*S. aureus* Strain	RFB	VCM	RFB	VCM
MRSA-C1	0.009 ± 0.004	1.226 ± 0.459	0.010 ± 0.006	>800
MRSA-C2	0.012 ± 0.001	1.875 ± 0.000	0.012 ± 0.002	>800

MIC: minimum inhibitory concentration; MBIC_50_: minimum biofilm inhibitory concentration able to inhibit 50% biofilm growth; RFB: rifabutin; VCM: vancomycin; MRSA: methicillin-resistant *S*. *aureus*. Results are expressed as mean ± SD (n = 3–4).

**Table 2 pharmaceuticals-17-00470-t002:** Physicochemical properties of RFB liposomal formulations.

Formulation	Lipid Composition (Molar Ratio)	Loading Capacity (μg RFB/µmol Lipid)	I.E. (%)	Ø (nm) (PdI)	ζ Pot. (mV)
RFB-LIP1	DMPC:DMPG:DSPE-PEG (65:30:5)	43 ± 5	55 ± 9	108 ± 8 (0.086 ± 0.046)	−5 ± 1
RFB-LIP2	DPPC:DPPG:DSPE-PEG (65:30:5)	36 ± 4	43 ± 7	110 ± 6 (0.078 ± 0.020)	−5 ± 1
RFB-LIP3	DMPC:DMPG (80:20)	36 ± 2	54 ± 3	108 ± 9 (0.059 ± 0.014)	−14 ± 1
RFB-LIP4	DPPC:DPPG (80:20)	37 ± 4	47 ± 8	116 ± 4 (0.089 ± 0.020)	−15 ± 1
Unloaded-LIP1	DMPC:DMPG:DSPE-PEG (65:30:5)	N.A.	N.A.	110 ± 11 (0.052 ± 0.025)	−5 ± 1
Unloaded-LIP2	DPPC:DPPG:DSPE-PEG (65:30:5)	N.A.	N.A.	114 ± 7 (0.100 ± 0.024)	−5 ± 1
Unloaded-LIP3	DMPC:DMPG (80:20)	N.A.	N.A.	108 ± 12 (0.097 ± 0.009)	−15 ± 2
Unloaded-LIP4	DPPC:DPPG (80:20)	N.A.	N.A.	121 ± 3 (0.062 ± 0.001)	−15 ± 3

Initial lipid concentration [Lip]i: 30 μmol/mL; initial RFB concentration: 100 nmol/μmol of lipid; loading capacity: RFB per lipid ratio (μg/µmol) in final liposomes; I.E.: incorporation efficiency; Ø: mean size of liposomes; PdI: polydispersity index; ζ pot.: zeta potential; DMPC: dimyristoyl phosphatidyl choline; DMPG: dimyristoyl phosphatidyl glycerol; DPPC: dipalmitoyl phosphatidyl choline; DPPG: dipalmitoyl phosphatidyl glycerol; DSPE-PEG: distearoyl phosphatidyl ethanolamine covalently linked to poly(ethylene)glycol 2000; N.A.: not applicable. Results are expressed as mean ± SD (n = 3–10).

**Table 3 pharmaceuticals-17-00470-t003:** Susceptibility of planktonic and biofilm MRSA-C1 to tested RFB formulations.

Formulation	MIC (μg/mL)	MBIC_50_ (μg/mL)
Free RFB	0.009 ± 0.004 ^a^	0.010 ± 0.006
RFB-LIP1	0.013 ^a^	0.008 ± 0.001
RFB-LIP2	0.013 ^a^	0.008 ± 0.001

^a^ Turbidimetry; MIC: minimum inhibitory concentration; MBIC_50_: minimum biofilm inhibitory concentration able to inhibit 50% biofilm growth; RFB: rifabutin; MRSA: methicillin-resistant *S*. *aureus*. RFB-LIP1: DMPC:DMPG:DSPE-PEG; RFB-LIP2: DPPC:DPPG:DSPE-PEG. Results are expressed as mean ± SD (n = 3).

## Data Availability

Data will be made available on request.

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
