# Peer review of "Liposomal Rifabutin—A Promising Antibiotic Repurposing Strategy against Methicillin-Resistant Staphylococcus aureus Infections"

_pharmaceuticals, 2024, doi:10.3390/ph17040470_

Round 1

Reviewer 1 Report

Comments and Suggestions for Authors

The manuscript entitled “Liposomal rifabutin - a promising antibiotic repurposing strategy 2 against methicillin-resistant Staphylococcus aureus infections” demonstrate the capability of liposomal rifabutin strategy to regulate the Staphylococcus aureus infection. However, the quality of the manuscript may be further strengthened by addressing the following queries:

1.      In the abstract section, Authors have mentioned the range of MIC and MBIC50. However, authors should also highlight the effective data values of RFB in free and liposomal forms towards the targeted bacteria control and their superiority/viability of liposomal forms over the free forms.

2.      In the line 29-32 of the abstract section, the flow of information should be improved to highlight the outcome of result findings.

3.      In the introduction section, the mechanism of used antibiotics against the target microbes may be briefly discussed.

4.      In the figure 3, the trends of bacterial viability against RFB/VCM are appearing the same. Authors should discuss why the antibiotic dose above the 0.207 microgram per milliliter is giving the antagonistic results.

5.      In Table 1 and Table 2, the free form of antibiotic is studied. Authors should discuss the effect of liposomal loaded form for the better understanding and efficacy of the drug towards targeted bacterium. Tables 1 and 2 may be clubbed together.

6.      In Table 3, authors should add one last column where the data values for the effectiveness of the bacterial viability may be provided.

7.      Table 4 provides the comparative data of the free and immobilized/liposomal form. However, the efficiency of the free form is better than the liposomal form. Authors should discuss the other associated benefits of using the liposomal based technology over the free form to strengthen the present main finding.

Author Response

Reviewer 1

The manuscript entitled “Liposomal rifabutin - a promising antibiotic repurposing strategy 2 against methicillin-resistant Staphylococcus aureus infections” demonstrate the capability of liposomal rifabutin strategy to regulate the Staphylococcus aureus infection. However, the quality of the manuscript may be further strengthened by addressing the following queries:

  1. In the abstract section, Authors have mentioned the range of MIC and MBIC50. However, authors should also highlight the effective data values of RFB in free and liposomal forms towards the targeted bacteria control and their superiority/viability of liposomal forms over the free forms.

Reply: We thank the Reviewer for the suggestion. We have included more detailed information in the Abstract. The following paragraph was included:

“RFB in free and liposomal forms displayed high antibacterial activity, with similar potency between tested formulations. In MRSA-C1, minimal inhibitory concentrations (MIC) for Free RFB and liposomal RFB were 0.009 and 0.013 μg/mL, respectively. Minimum biofilm inhibitory concentration able to inhibit 50% biofilm growth (MBIC50) for Free RFB and liposomal RFB against MRSA-C1 were 0.012 and 0.008 μg/mL, respectively.”

  1. In the line 29-32 of the abstract section, the flow of information should be improved to highlight the outcome of result findings.

Reply: We appreciate the useful suggestion from the Reviewer. Accordingly, we have revised that section in the Abstract as follows:

“The in vivo results demonstrated a significant reduction of bacterial burden and growth index in major organs of mice treated with RFB formulations, as compared to Control and VCM (positive control) groups. Furthermore, VCM therapeutic dose was two-fold higher than the one used RFB formulations, reinforcing the therapeutic potency of the proposed strategy.”

  1. In the introduction section, the mechanism of used antibiotics against the target microbes may be briefly discussed.

Reply: We thank the Reviewer’s recommendation. We have added additional information regarding RFB and VCM mechanisms of action:

“VCM shows activity against most Gram-positive cocci and bacilli. It acts by interrupting cell wall synthesis in dividing bacteria by specifically inhibiting the incorporation of murein monomers into the peptidoglycan chain [11].”

“Rifamycins are active against mycobacterium, Gram-positive bacteria, and, to a lower extent, Gram-negative bacteria. These antibiotics bind to the prokaryotic DNA-dependent RNA polymerase, suppressing transcription and protein synthesis [24–26]. As this mechanism of action is independent of bacterial division, bacteria populations with low metabolic activity, such as those in biofilms, are also susceptible [25].”

  1. In the figure 3, the trends of bacterial viability against RFB/VCM are appearing the same. Authors should discuss why the antibiotic dose above the 0.207 microgram per milliliter is giving the antagonistic results.

Reply: Thank you for your comment. In Figure 3 (Figure 4 in the revised manuscript) are shown the results of bacterial cell viability, in percentage, of two clinical strains after incubation with tested antibiotics in the free form. The observed antibacterial activity is concentration-dependent. Concentrations above or equal to 0.207 μg/ml resulted in similar reduction of bacterial cell viability, and this was confirmed by statistical analysis that showed no statistically significant differences among concentrations equal or above 0.207 μg/ml. Therefore, the authors must disagree with the Reviewer statement that an antagonistic effect is observed for concentrations >0.207 μg/ml since the antibacterial activity is maintained.

  1. In Table 1 and Table 2, the free form of antibiotic is studied. Authors should discuss the effect of liposomal loaded form for the better understanding and efficacy of the drug towards targeted bacterium. Tables 1 and 2 may be clubbed together.

Reply: We appreciate the Reviewer’s comments and suggestions. In the revised manuscript, the authors have put MIC and MBIC50 of the free form results in a single table (now designated as Table 1).

Addressing the Reviewer’s question regarding the discussion of liposome-loaded antibiotic, both the MIC and MBIC50 values are very similar between free RFB and liposomal RFB. This is an important finding since it confirms that the antibacterial activity of RFB was retained after incorporation in liposomes. Another important fact is that the antibacterial effect of liposomal RFB was solely due to the presence of the antibiotic, as demonstrated by the lack of effect of unloaded liposomes (Figures 5 and 6 of the revised manuscript). The remarkable antibacterial activity of RFB is confirmed by these in vitro results that showed no statistically significant differences between free and liposomal RFB. Nevertheless, the use of liposomes usually aims to improve the therapeutic effectiveness and/or safety of associated compounds in vivo. In this sense and considering the Reviewer’s suggestion, the authors have added additional discussion at the end of Results and Discussion section regarding the advantages of liposomes for antibiotic delivery:

“As aforementioned, the use of liposomes aims to improve the efficacy and/or the safety of associated drugs. Antibiotics in the free form may display low bioavailability and concentration at infected sites, a consequence of their unfavorable biodistribution profile [18,31]. Only a fraction of the originally administered antibiotic dose reaches the target, and this may influence the emergence of resistance [18,31]. The low concentration of antibiotic at infected sites requires frequent administrations of high doses in order to exert a therapeutic effect being in the origin of toxic side effects [31]. To overcome these hurdles, liposomes offer many advantages as antibiotic nanocarriers. These lipid-based nanosystems may reduce dose and frequency of administrations, prevent enzymatic degradation, avoid drug efflux mechanisms, and overcome outer membranes [18,31,68,69]. In the case of RFB, previous studies in a murine model of disseminated Mycobacterium tuberculosis infection demonstrated a significantly superior antimycobacterial effect of RFB-loaded liposomes in the spleen and liver, compared to free antibiotic [28]. Moreover, compared with free RFB, lung inflammation was reduced in mice treated with liposomal RFB, which indicated a better safety profile [28].

Importantly, liposomes are advantageous for the eradication of biofilm-organized bacteria, such as those formed in orthopedic implants and medical devices, promoting the penetration within the biofilm and release of loaded antibacterial agent [18,31,68]. Moreover, different lipid compositions can be designed to maximize the preferential accumulation of liposomal formulations at diseased sites, including the functionalization of surface with specific moieties for a targeted approach [70,71]. Noteworthy, in the present work, only one dose and one treatment schedule were tested. Different therapeutic regimens could be further assessed to optimize the therapeutic index of RFB formulations.”

  1. In Table 3, authors should add one last column where the data values for the effectiveness of the bacterial viability may be provided.

Reply: We acknowledge the Reviewer’s observations. However, Table 2 (revised manuscript) contains only data regarding the physicochemical properties of developed liposomes, accounting for the different lipid compositions and the presence/absence of the antibiotic RFB. For the in vitro evaluation in MRSA, only liposomes with long blood circulating properties (containing DSPE-PEG) were selected, considering the subsequent in vivo application. In Figures 5 and 6 and in Table 3 (revised manuscript) are depicted the obtained data for the two RFB-loaded liposomes (RFB-LIP1 and RFB-LIP2), as well as the respective unloaded liposomes. MIC and MBIC50 values were determined and compared with the free antibiotic. The authors believe that, with the current format, the flow of information is better understood by the readers.

  1. Table 4 provides the comparative data of the free and immobilized/liposomal form. However, the efficiency of the free form is better than the liposomal form. Authors should discuss the other associated benefits of using the liposomal based technology over the free form to strengthen the present main finding.

Reply: The authors thank the Reviewer’s comments. The information in Table 3 (revised manuscript) is relative to the susceptibility of planktonic and biofilm bacteria to RFB in free and liposomal forms. No statistically significant differences were obtained for tested formulations, indicating a similar antibacterial activity. Nevertheless, the benefits of liposomes as antibiotic nanocarriers are more relevant in the in vivo setting as a result of a preferential accumulation at diseased sites.

Reviewer 2 Report

Comments and Suggestions for Authors

The topic of this work is interesting and the content is informative. There is a big problem, the literature survey cited the very old works and I can rarely find a recent work in the reference list. I can advice the authors to revise their work including recent works in the reference list. Also, compare the advantages of this work with those of recent works. 

Author Response

Reviewer 2

The topic of this work is interesting and the content is informative. There is a big problem, the literature survey cited the very old works and I can rarely find a recent work in the reference list. I can advice the authors to revise their work including recent works in the reference list. Also, compare the advantages of this work with those of recent works. 

Reply: We thank the Reviewer for the pertinent comments. Additional recent papers that focus on liposomal delivery of antibiotics to MRSA infections have been included in the revised version of the manuscript. At the end of section 2.4, new information was added as follows:

“Recent studies with antibiotic-loaded liposomes confirm the potential of this strategy against S. aureus. A work from Ferreira et al (2021) involved the design of RFB liposomes with different lipid compositions, and the assessment of in vitro antibacterial activity against planktonic and biofilm MSSA [20]. Compared to Ferreira et al (2021), in our study we focused on the design of RFB-loaded liposomes including DSPE-PEG and tested the antibacterial effectiveness using clinical MRSA strains for in vitro and in vivo assays.

Researchers have also explored liposomal formulations of different antibiotics. For instance, Rani et al (2022) have designed liposomes co-loading VCM and daptomycin, with an external surface mimicking the red blood cells membrane [62]. The authors only evaluated the antibacterial activity against planktonic bacteria, while our study also presents biofilm results. In addition, we performed the in vivo therapeutic evaluation in MRSA systemic models, while Rani et al (2022) conducted biodistribution and safety evaluation in healthy rats [62]. Moreover, S. aureus strains resistant to VCM and daptomycin are emerging [63], contrary to RFB that is currently applied to treat mycobacteria infections and, in the present work, the main goal was to validate the repurpose of this antibiotic towards S. aureus infection.

Ashar and colleagues (2023) have developed temperature-sensitive liposomes loading ciprofloxacin [64]. This liposomal formulation, with a mean size of 184 nm, was tested

 in a rat model of implant-associated MRSA biofilm osteomyelitis. After intravenous administration of liposomes, high-intensity focused ultrasound was used to promote the antibiotic release at the site of bone infection [64]. While Ashar et al (2023) only assessed biofilm infection targeting, in the present study smaller liposomes (around 100 nm) loading RFB were designed and these nanoformulations were shown to be effective against both planktonic and biofilm MRSA, without the need to resort to additional external stimulus and/or equipment.

The research team of Makhathini and collaborators (2019) developed VCM-loaded pH-responsive liposomes containing newly synthetized fatty acid-based zwitterionic lipids [65]. The different liposomal formulations were tested in vitro against planktonic MRSA and the proof-of-concept was conducted in a murine model of skin infection being the tested formulations locally administered into the infection site [65]. In the current work, systemic models of MRSA infection were established, representing a more advanced infection status. Furthermore, for the preparation of liposomal formulations, we used phospholipids that are currently present in several approved nanoformulations [66], while Makhathini et al (2019) synthetized new lipids to design pH-sensitive liposomes [65]. Overall, these recent works with liposomes as antibiotic nanocarriers highlight the investment in this lipid-based nanosystem. Liposomes prove to be a versatile and advantageous tool for the development of more effective and safe therapeutic approaches against S. aureus.”

[20]  M. Ferreira, S.N. Pinto, F. Aires-da-Silva, A. Bettencourt, S.I. Aguiar, M.M. Gaspar, Liposomes as a nanoplatform to improve the delivery of antibiotics into Staphylococcus aureus biofilms, Pharmaceutics. 13 (2021) 321. https://doi.org/10.3390/pharmaceutics13030321.

[62]  N.N.I.M. Rani, X.Y. Chen, Z.M. Al-Zubaidi, H. Azhari, T.M.N. Khaitir, P.Y. Ng, F. Buang, G.C. Tan, Y.P. Wong, M.M. Said, A.M. Butt, A.A. Hamid, M.C.I.M. Amin, Surface-engineered liposomes for dual-drug delivery targeting strategy against methicillin-resistant Staphylococcus aureus (MRSA), Asian J. Pharm. Sci. 17 (2022) 102–119. https://doi.org/10.1016/j.ajps.2021.11.004.

[63]  M. Roch, P. Gagetti, J. Davis, P. Ceriana, L. Errecalde, A. Corso, A.E. Rosato, Daptomycin Resistance in Clinical MRSA Strains Is Associated with a High Biological Fitness Cost, Front. Microbiol. 8 (2017) 2303. https://doi.org/10.3389/fmicb.2017.02303.

[64]  H. Ashar, A. Singh, K. Ektate, S. More, A. Ranjan, Treating methicillin-resistant Staphylococcus aureus (MRSA) bone infection with focused ultrasound combined thermally sensitive liposomes, Int. J. Hyperth. 40 (2023) 2211278. https://doi.org/10.1080/02656736.2023.2211278.

[65]  S.S. Makhathini, R.S. Kalhapure, M. Jadhav, A.Y. Waddad, R. Gannimani, C.A. Omolo, S. Rambharose, C. Mocktar, T. Govender, Novel two-chain fatty acid-based lipids for development of vancomycin pH-responsive liposomes against Staphylococcus aureus and methicillin-resistant Staphylococcus aureus (MRSA), J. Drug Target. 27 (2019) 1094–1107. https://doi.org/10.1080/1061186X.2019.1599380.

[66] H. Luiz, J. Oliveira Pinho, M.M. Gaspar, Advancing Medicine with Lipid-Based Nanosystems—The Successful Case of Liposomes, Biomedicines. 11 (2023) 435. https://doi.org/10.3390/biomedicines11020435.

Reviewer 3 Report

Comments and Suggestions for Authors

In this study, the authors developed a liposomal system loaded with Rifabutin (RFB) and studied their efficacy against drug-resistant MRSA. The liposomes preserved the antibacterial activity of RFB and it is similar to free RFB. Also, the authors performed in vivo studies to support their findings. This is an interesting study, and the experiments are well designed. The authors are suggested to address following comments before acceptance.

1. The authors are suggested to perform morphological studies such as TEM to see the morphology of liposomal system.

2. Statistical analysis needs to be performed to all the figures wherever is necessary.

3. Why the liposomes accumulate at infected site. What are the interactions involved?

4. Schematic diagram will be helpful to understand this study.

5. What is the cytotoxicity of these formulations?

6. The therapeutic efficacy of free RFB and RFB loaded liposomes seems similar in both in vitro and in vivo. So, advantages of carrier need to be exposed.

7. What is the release profile of drug from liposomes.

Author Response

Reviewer 3

In this study, the authors developed a liposomal system loaded with Rifabutin (RFB) and studied their efficacy against drug-resistant MRSA. The liposomes preserved the antibacterial activity of RFB and it is similar to free RFB. Also, the authors performed in vivo studies to support their findings. This is an interesting study, and the experiments are well designed. The authors are suggested to address following comments before acceptance.

  1. The authors are suggested to perform morphological studies such as TEM to see the morphology of liposomal system.

Reply: The authors appreciate the Reviewer’s suggestion. TEM is a powerful tool to provide information of liposome morphology and architecture, including the number of lipid bilayers. However, this technique is also limited as it requires a high vacuum that may result in liposome aggregation during their preparation for imaging. In addition, TEM suffers from limited sampling, and so data reported are not necessarily representative of all the NPs in one formulation, being also a long processing and time cost methodology. On the other hand, DLS is the most commonly used technique for the particle size analyses of nanoparticles such as liposomes. In addition, to the average mean size the polydispersity index (PDI) is an important characteristic that is possible to obtain by DLS, giving information about the homogeneity of a liposomal suspension that ranges from 0 to 1 from a monodisperse to polydisperse suspension, respectively. In addition, DLS is a very rapid and cheap method that allows the determination of liposomal suspensions not only in a final sample but also during liposomes preparation and, consequently, it is possible to produce liposomes with the appropriate mean size and PDI. In present work, our main objective was to achieve liposomes with a PDI below 0.1.

Although the combination of DLS and TEM serves as a comprehensive approach to size and distribution measurements that should better satisfy FDA/EMA requirements, we are not yet at that stage of development. Nevertheless, the authors will consider TEM analysis in future experiments to further characterize the developed formulations.

  1. Statistical analysis needs to be performed to all the figures wherever is necessary.

Reply: We thank the Reviewer for this suggestion. The authors have performed additional statistical analysis that we considered to be relevant. In the revised version of the manuscript, additional statistical analysis was included Figure 3, Figure 9 and Figure 10. The respective statistical information was added in the captions.

  1. Why the liposomes accumulate at infected site. What are the interactions involved?

Reply: The authors appreciate the Reviewer’s questions. Liposomes versatility in terms of composition and physicochemical properties allow researchers to design appropriate liposomal formulations according to the desired target tissue/organ. In the case of infections mainly localized in liver and spleen, the accumulation of liposomes in these organs usually does not require the inclusion of DSPE-PEG [1-3]. This has been demonstrated in previous studies using murine models of Mycobacterium avium, Mycobacterium tuberculosis and Leishmania infantum infections [1,2]. In our case, the most affected organs were the liver, spleen, lungs and this was the reason why we selected nanoformulations including DSPE-PEG in the lipid composition. When other major organs, besides liver and spleen, are affected, namely the lungs, liposomes are required to display long blood circulating times, thus allowing the extravasation for longer times to infected sites [2,3]. Furthermore, we aim to promote the interaction of liposomes with target bacteria, and the presence of DSPE-PEG offers advantages. Since bacteria display a negative surface charge and liposomes with DSPE-PEG present neutral surface charge, a high level of interaction between liposomes and bacteria is achieved ([10,11] and reviewed in [4]).

[1]    M.M. Gaspar, S. Calado, J. Pereira, H. Ferronha, I. Correia, H. Castro, A.M. Tomás, M.E.M. Cruz, Targeted delivery of paromomycin in murine infectious diseases through association to nano lipid systems, Nanomedicine Nanotechnology, Biol. Med. 11 (2015) 1851–1860. https://doi.org/10.1016/j.nano.2015.06.008.

[2]    M.M. Gaspar, A. Cruz, A.F. Penha, J. Reymão, A.C. Sousa, C.V. Eleutério, S.A. Domingues, A.G. Fraga, A.L. Filho, M.E.M. Cruz, J. Pedrosa, Rifabutin encapsulated in liposomes exhibits increased therapeutic activity in a model of disseminated tuberculosis, Int. J. Antimicrob. Agents. 31 (2008) 37–45. https://doi.org/10.1016/j.ijantimicag.2007.08.008.

[3]    M. Gaspar, A. Cruz, A. Fraga, A. Castro, M. Cruz, J. Pedrosa, Developments on drug delivery systems for the treatment of mycobacterial infections, Curr. Top. Med. Chem. 8 (2008) 579–591. https://doi.org/10.2174/156802608783955629.

[4]    M. Ferreira, S. Aguiar, A. Bettencourt, M.M. Gaspar, Lipid-based nanosystems for targeting bone implant-associated infections: Current approaches and future endeavors, Drug Deliv. Transl. Res. 11 (2021) 72–85. https://doi.org/10.1007/s13346-020-00791-8.

[10]  S. Pereira, R.S. Santos, L. Moreira, N. Guimarães, M. Gomes, H. Zhang, K. Remaut, K. Braeckmans, S. De Smedt, N.F. Azevedo, Lipoplexes to Deliver Oligonucleotides in Gram-Positive and Gram-Negative Bacteria: Towards Treatment of Blood Infections, Pharmaceutics. 13 (2021) 989. https://doi.org/10.3390/pharmaceutics13070989.

[11]  L. Moreira, N.M. Guimarães, S. Pereira, R.S. Santos, J.A. Loureiro, M.C. Pereira, N.F. Azevedo, Liposome Delivery of Nucleic Acids in Bacteria: Toward In Vivo Labeling of Human Microbiota, ACS Infect. Dis. 8 (2022) 1218–1230. https://doi.org/10.1021/acsinfecdis.1c00601.

  1. Schematic diagram will be helpful to understand this study.

Reply: The authors appreciate the valuable suggestion. We have included, at the end of the Introduction, a new figure (Figure 1) that represents the disease in focus, the aim of the study and the experimental design:

“Figure 1. Schematic representation of the experimental design for the development of RFB-loaded liposomes as a therapeutic strategy against MRSA infections.”

  1. What is the cytotoxicity of these formulations?

Reply: The present work aimed to validate the antibacterial effect of RFB both in free and liposomal forms against clinical MRSA strains. We understand that for further development of the tested RFB liposomal formulations several in vitro requirements are needed, including the cytotoxicity effect of the selected liposomal formulations in different cell lines such as THP1, J774 or healthy liver cell lines, among others. In previous work performed by our research group [1], cell viability of human MG-63 osteoblasts and mouse L929 fibroblasts were assessed after incubation with unloaded and RFB loaded in liposomes. In that work, three different lipid compositions were tested Both cell lines were incubated for 24 h at lipid concentrations that ranged from 0.5 to 5.0 µmol/mL. This incubation and lipid concentrations did not compromise cell viability, particularly for the two negatively-charged lipid compositions, that presented a cell viability superior to 80%. In our research work, the maximum lipid concentrations tested were <0.006 µmol/mL. Considering that in the present work much lower lipid concentrations were used, it is reasonable to expect that selected lipid compositions will not induce cytotoxic effect. Moreover, the phospholipids included in the tested lipid compositions are present in several liposomal formulations in clinical use [2].

[1] M. Ferreira, S.N. Pinto, F. Aires-da-Silva, A. Bettencourt, S.I. Aguiar, M.M. Gaspar, Liposomes as a nanoplatform to improve the delivery of antibiotics into Staphylococcus aureus biofilms, Pharmaceutics. 13 (2021) 321. https://doi.org/10.3390/pharmaceutics13030321.

[2] H. Luiz, J. Oliveira Pinho, M.M. Gaspar, Advancing Medicine with Lipid-Based Nanosystems—The Successful Case of Liposomes, Biomedicines. 11 (2023) 435. https://doi.org/10.3390/biomedicines11020435.

  1. The therapeutic efficacy of free RFB and RFB loaded liposomes seems similar in both in vitro and in vivo. So, advantages of carrier need to be exposed.

Reply: We appreciate the Reviewers comment and suggestion. In vitro results demonstrated a similar antibacterial activity of RFB in free and liposomal forms. This is important since it shows that the antibacterial potency of RFB was preserved after association to liposomes. In vivo, two murine models of MRSA systemic infection were evaluated. In the first and second models, inocula of 1.4x10^9 CFU/ml and 3.4x10^8 CFU/ml were used for infection induction, respectively. Although in the In Vivo Assay 2 (lower MRSA inoculum; Supplementary Figure S4) free and liposomal RFB (RFB-LIP1 and RFB-LIP2) showed equivalent therapeutic effects, in the murine model induced with a 4-fold higher MRSA inoculum (In Vivo Assay 1; Figure 9), the superior therapeutic effectiveness of RFB-LIP2 was evidenced. Here, RFB-LIP2 achieved a significant antibacterial effect in all tested organs, compared to Control and Free RFB. As infection sites may be exposed to subtherapeutic concentrations of antibiotics when in the free form, these results are of extreme importance since, despite the higher inoculum, liposomal RFB was able to overcome the limitations of Free RFB, exerting a significant therapeutic effect, with no adverse events. In the revised version of the manuscript, in section 2.6, this discussion has been added as follows:

“Furthermore, the superior therapeutic effectiveness of RFB-LIP2 was evidenced in the model that was induced with a 4-fold higher MRSA inoculum, achieving a significant antibacterial effect in all tested organs, compared to Control and Free RFB (Figure 9). These results are of extreme importance given that infection sites may be exposed to subtherapeutic concentrations of antibiotics when in the free form. This may contribute to the emergence of resistance and cause severe systemic toxicity [18,31]. Despite this situation being aggravated by higher inoculum, liposomal RFB was able to overcome these challenges and exert a significant therapeutic effect, with no adverse events.”

  1. What is the release profile of drug from liposomes.

Reply: We appreciate the Reviewer’s question. The authors have performed stability studies of RFB-LIP1 and RFB-LIP2 after storage at 4 ºC, during 7 days; however, these results were not included in the submitted manuscript. Following the Reviewer’s helpful suggestion, the authors have added, in the revised Manuscript and Supplementary Materials, the stability results of developed PEGylated nanoformulations in terms of mean size, PdI, and percentage of associated RFB (Supplementary Materials, Figure S2). In Section 2.3., the new information was added:

“The stability of DSPE-PEG-containing liposomal suspensions, RFB-LIP1 and RFB-LIP2, was assessed after a storage period of 7 days, at 4 °C (Supplementary Methods and Figure S2). Obtained data indicated that the more fluid lipid composition, DMPC:DMPG:DSPE-PEG (RFB-LIP1), displayed a higher % of release of the antibiotic compared to the more rigid one DPPC:DPPG:DSPE-PEG (RFB-LIP2). RFB-LIP1 and RFB-LIP2 showed a RFB release of 35 and 16%, respectively, after 7 days of storage.

Reviewer 4 Report

Comments and Suggestions for Authors

 "Liposomal rifabutin - a promising antibiotic repurposing strategy against methicillin-resistant Staphylococcus aureus infection" by M.M. Gaspar and colleagues presents a promising strategy for using rifabutin-loaded liposomes versus methicillin-resistant Staphylococcus aureus.

The manuscript is well-written and designed and includes relevant data in the field.I recommend acceptance of the manuscript for publication after the Authors have responded to the comments below:

1)      Figure 1 - why these concentration ranges were chosen for the study (RFB and VCM concentrations from 0.0016-0.414 μg/ml and 0.014-3.750 μg/ml, respectively)?

2)      On what basis were the compositions of the solutions used to prepare the liposomes selected? Is there any literature background to justify the choice of each?

3)      Please provide PDI with greater accuracy in Table 3.

4)      Please include in the Supplementary Data examples of the plots of the liposome size distribution by number and intensity for the selected loaded and unloaded liposomes, together with a table displayed by the ZetaSizer apparatus in which the liposome sizes and PDI will be placed.

5)      Generally, a large positive or negative value of the zeta potential (lower than -30 mV and higher than +30 mV) indicates physical stability due to the electrostatic repulsion of individual particles. On the other hand, a small value of the electrokinetic potential may cause aggregation of particles due to the van der Waals forces. The electrokinetic potential values obtained by the Authors are small and suggest poor stability of the systems obtained. Therefore, it seems necessary to carry out a study to check the value of mean size, zeta potential, and incorporation efficiency in time. Please make such measurements at least over 2 or 3 days (a 2 or 3-month study would be desirable).

6)      278-279 lines – Please explain how the presence of PEG in the liposome formulation affects liposome activity.

Author Response

Reviewer 4

"Liposomal rifabutin - a promising antibiotic repurposing strategy against methicillin-resistant Staphylococcus aureus infection" by M.M. Gaspar and colleagues presents a promising strategy for using rifabutin-loaded liposomes versus methicillin-resistant Staphylococcus aureus.

The manuscript is well-written and designed and includes relevant data in the field. I recommend acceptance of the manuscript for publication after the Authors have responded to the comments below:

1) Figure 1 - why these concentration ranges were chosen for the study (RFB and VCM concentrations from 0.0016-0.414 μg/ml and 0.014-3.750 μg/ml, respectively)?

Reply: We appreciate the Reviewer’s comments on the submitted manuscript. In these in vitro assays, the goal was to assess the minimum concentration at each one of the antibiotics inhibits bacterial growth. The tested concentrations correspond to 2-fold dilutions. We have tested a wide range of concentrations for each antibiotic in order to have, at least, two to three concentrations above the minimum inhibitory concentration (MIC) where the absence of bacterial growth is clearly confirmed.

2) On what basis were the compositions of the solutions used to prepare the liposomes selected? Is there any literature background to justify the choice of each?

Reply: We thank the Reviewer for the questions. The lipid compositions tested by the authors in this work were based on previous studies. On the one hand, considering that we want to reach infected sites in vivo, besides liver and spleen, we have included DSPE-PEG in the lipid composition to promote long blood circulating properties. On the other hand, these type of lipid compositions, as well as those negatively charged, have been successfully employed for delivering loaded compounds to infected sites, namely mycobacterial, bacterial, and parasitic infections with success [1–6]. Previous results with some of these lipid compositions have been obtained in MSSA in vitro assays (DMPC:DMPG and DPPC:DPPG) [7]; in a Mycobacterium avium model of infection (DPPC:DPPG) [3]; in models of disseminated M. tuberculosis infection (DMPC:DMPG, DPPC:DPPG and HPC:DSPE-PEG) [5]; in Leishmania infantum [5,6] and M. avium models [3,5] (DMPC:DMPG, DPPC:DPPG, DMPC:DMPG:DSPE-PEG and DPPC:DPPG:DSPE-PEG). These previous results showed the therapeutic effectiveness of developed liposomal formulations and their favorable biodistribution profile in major organs, such as the spleen, liver and lungs, which are also some of the most affected ones in MRSA systemic infection. Following these data, the authors selected the lipid compositions DMPC:DMPG and DPPC:DPPG and included DSPE-PEG considering the in vivo applications of the liposomal formulations (DMPC:DMPG:DSPE-PEG and DPPC:DPPG:DSPE-PEG). The presence of DSPE-PEG at liposome surface aimed to increase the blood residence time to further enhance the passive accumulation of liposomes at infected sites and, consequently, increase local concentration of RFB and enhance the therapeutic effect.

In section 2.6 of the revised manuscript, additional information regarding the rational for the selection of lipid compositions was added.

[1]        M. Ferreira, S. Aguiar, A. Bettencourt, M.M. Gaspar, Lipid-based nanosystems for targeting bone implant-associated infections: Current approaches and future endeavors, Drug Deliv. Transl. Res. 11 (2021) 72–85. https://doi.org/10.1007/s13346-020-00791-8.

[2]        M. Ferreira, M. Ogren, J.N.R. Dias, M. Silva, S. Gil, L. Tavares, F. Aires-da-Silva, M.M. Gaspar, S.I. Aguiar, Liposomes as antibiotic delivery systems: A promising nanotechnological strategy against antimicrobial resistance, Molecules. 26 (2021) 2047. https://doi.org/10.3390/molecules26072047.

[3]        M.M. Gaspar, S. Neves, F. Portaels, J. Pedrosa, M.T. Silva, M.E.M. Cruz, Therapeutic efficacy of liposomal rifabutin in a Mycobacterium avium model of infection, Antimicrob. Agents Chemother. 44 (2000) 2424–2430. https://doi.org/10.1128/AAC.44.9.2424-2430.2000.

[4]        M. Gaspar, A. Cruz, A. Fraga, A. Castro, M. Cruz, J. Pedrosa, Developments on drug delivery systems for the treatment of mycobacterial infections, Curr. Top. Med. Chem. 8 (2008) 579–591. https://doi.org/10.2174/156802608783955629.

[5]        M.M. Gaspar, S. Calado, J. Pereira, H. Ferronha, I. Correia, H. Castro, A.M. Tomás, M.E.M. Cruz, Targeted delivery of paromomycin in murine infectious diseases through association to nano lipid systems, Nanomedicine Nanotechnology, Biol. Med. 11 (2015) 1851–1860. https://doi.org/10.1016/j.nano.2015.06.008.

[6]        R.M. Lopes, J. Pereira, M.A. Esteves, M.M. Gaspar, M. Carvalheiro, C. V Eleutério, L. Gonçalves, A. Jiménez-Ruiz, A.J. Almeida, M.E.M. Cruz, Lipid-based nanoformulations of trifluralin analogs in the management of Leishmania infantum infections, Nanomedicine. 11 (2016) 153–170. https://doi.org/10.2217/nnm.15.190.

[7]        M. Ferreira, S.N. Pinto, F. Aires-da-Silva, A. Bettencourt, S.I. Aguiar, M.M. Gaspar, Liposomes as a nanoplatform to improve the delivery of antibiotics into Staphylococcus aureus biofilms, Pharmaceutics. 13 (2021) 321. https://doi.org/10.3390/pharmaceutics13030321.

3) Please provide PDI with greater accuracy in Table 3.

Reply: We appreciate the Reviewer’s input. Accordingly, in the revised version of the manuscript, the authors have added in Table 2 (revised version of the manuscript) detailed values of PdI, including the standard deviation.

4) Please include in the Supplementary Data examples of the plots of the liposome size distribution by number and intensity for the selected loaded and unloaded liposomes, together with a table displayed by the ZetaSizer apparatus in which the liposome sizes and PDI will be placed.

Reply: We thank the Reviewer’s suggestion. In the revised version of the Supplementary Materials, the authors have included Figure S1 that displays representative DLS plots recorded by the MalvernS for RFB-loaded and unloaded LIP1 (DMPC:DMPG:DSPE-PEG) and LIP2 (DPPC:DPPG:DSPE-PEG), the two more relevant lipid compositions used in both in vitro and in vivo studies.

5) Generally, a large positive or negative value of the zeta potential (lower than -30 mV and higher than +30 mV) indicates physical stability due to the electrostatic repulsion of individual particles. On the other hand, a small value of the electrokinetic potential may cause aggregation of particles due to the van der Waals forces. The electrokinetic potential values obtained by the Authors are small and suggest poor stability of the systems obtained. Therefore, it seems necessary to carry out a study to check the value of mean size, zeta potential, and incorporation efficiency in time. Please make such measurements at least over 2 or 3 days (a 2 or 3-month study would be desirable).

Reply: We appreciate the Reviewer’s remarks and valuable suggestions. Although positively and negatively-charged formulations do not tend to aggregate due to electrostatic repulsion, their application is hindered due high toxicity (positively charged liposomes) and low interaction with the negatively charged bacteria cell wall, such is the case of liposomes with negative charge [1-3]. In the present study, although we designed liposomes with and without DSPE-PEG in the lipid composition, we focused on those with this polymer at the surface (LIP 1 and LIP2) since our goal was the in vivo application in murine models of MRSA infection. DSPE-PEG forms a protective hydrophilic layer at liposomes surface and often these are designated as “stealth” or “sterically stabilized” liposomes. This feature is of utmost importance for in vivo applications since DSPE-PEG reduces plasmatic protein adsorption to liposomes’ surface and uptake by the mononuclear phagocytic system, prolonging blood residence time and enhancing passive accumulation at diseased sites, namely infection and tumors [1-8]. In addition, the presence of DSPE-PEG at the liposomes’ surface also shields the surface charge, conferring a zeta potential close to neutrality (-10 mV to +10 mV), such is the case of our LIP1 and LIP2 nanoformulations. Despite the small zeta potential values observed for LIP-1 and LIP-2 (-5 mV), it is well-established that the PEG chains prevent nanoparticle aggregation and, thus, improve their stability. The repulsion between bilayers promoted by PEG is able to overcome the Van der Waals forces [4-9]. Furthermore, the fusion with bacteria cell envelope has been reported for liposomes with a neutral surface charge and/or DSPE-PEG coating ([10,11]and reviewed in [1]).

Regarding the stability of developed liposomal formulations, the authors had performed, but did not include in the manuscript, the physicochemical characterization of RFB-LIP1 and RFB-LIP2 over a period of 7 days storage, in buffer, at 4ºC. Following the Reviewer’s suggestion, the stability results of developed PEGylated nanoformulations were included in the revised Supplementary Materials (Figure S2).

[1]          M. Ferreira, S. Aguiar, A. Bettencourt, M.M. Gaspar, Lipid-based nanosystems for targeting bone implant-associated infections: Current approaches and future endeavors, Drug Deliv. Transl. Res. 11 (2021) 72–85. https://doi.org/10.1007/s13346-020-00791-8.

[2]          M. Ferreira, M. Ogren, J.N.R. Dias, M. Silva, S. Gil, L. Tavares, F. Aires-da-Silva, M.M. Gaspar, S.I. Aguiar, Liposomes as antibiotic delivery systems: A promising nanotechnological strategy against antimicrobial resistance, Molecules. 26 (2021) 2047. https://doi.org/10.3390/molecules26072047.

[3]          M. Ferreira, S.N. Pinto, F. Aires-da-Silva, A. Bettencourt, S.I. Aguiar, M.M. Gaspar, Liposomes as a nanoplatform to improve the delivery of antibiotics into Staphylococcus aureus biofilms, Pharmaceutics. 13 (2021) 321. https://doi.org/10.3390/pharmaceutics13030321.

[4]          B. Romberg, W.E. Hennink, G. Storm, Sheddable Coatings for Long-Circulating Nanoparticles, Pharm. Res. 25 (2008) 55–71. https://doi.org/10.1007/s11095-007-9348-7.

[5]          M.A. Phillips, M.L. Gran, N.A. Peppas, Targeted nanodelivery of drugs and diagnostics, Nano Today. 5 (2010) 143–159. https://doi.org/10.1016/j.nantod.2010.03.003.

[6]          L. Belfiore, D.N. Saunders, M. Ranson, K.J. Thurecht, G. Storm, K.L. Vine, Towards clinical translation of ligand-functionalized liposomes in targeted cancer therapy: Challenges and opportunities, J. Control. Release. 277 (2018) 1–13. https://doi.org/10.1016/j.jconrel.2018.02.040.

[7]          T.M. Allen, P.R. Cullis, Drug delivery systems: Entering the mainstream, Science (80-. ). 303 (2004) 1818–1822.

[8]          M.C. Smith, R.M. Crist, J.D. Clogston, S.E. McNeil, Zeta potential: a case study of cationic, anionic, and neutral liposomes, Anal. Bioanal. Chem. 409 (2017) 5779–5787. https://doi.org/10.1007/s00216-017-0527-z.

[9]          C. Allen, N. Dos Santos, R. Gallagher, G.N.C. Chiu, Y. Shu, W.M. Li, S.A. Johnstone, A.S. Janoff, L.D. Mayer, M.S. Webb, M.B. Bally, Controlling the Physical Behavior and Biological Performance of Liposome Formulations Through Use of Surface Grafted Poly(ethylene Glycol), Biosci. Rep. 22 (2002) 225–250. https://doi.org/10.1023/A:1020186505848.

[10]        S. Pereira, R.S. Santos, L. Moreira, N. Guimarães, M. Gomes, H. Zhang, K. Remaut, K. Braeckmans, S. De Smedt, N.F. Azevedo, Lipoplexes to Deliver Oligonucleotides in Gram-Positive and Gram-Negative Bacteria: Towards Treatment of Blood Infections, Pharmaceutics. 13 (2021) 989. https://doi.org/10.3390/pharmaceutics13070989.

[11]         L. Moreira, N.M. Guimarães, S. Pereira, R.S. Santos, J.A. Loureiro, M.C. Pereira, N.F. Azevedo, Liposome Delivery of Nucleic Acids in Bacteria: Toward In Vivo Labeling of Human Microbiota, ACS Infect. Dis. 8 (2022) 1218–1230. https://doi.org/10.1021/acsinfecdis.1c00601.

6) 278-279 lines – Please explain how the presence of PEG in the liposome formulation affects liposome activity.

Reply: We appreciate the Reviewer’s question. PEGylation is a known strategy to amplify antibiofilm activity of nanosystems and antibiotics. Several studies have reported that nanoparticulated systems with their surface decorated with PEG rapidly and effectively penetrate within S. aureus biofilm [1]. It is hypothesized that PEG coating diminishes the binding affinity to biofilm matrix which, in turn, amplifies the penetration into the interior layers of biofilm [2–5]. This has also been described for PEGylated antibiotics [6]. The PEG coating reduces electrostatic interactions with mucus and biofilm matrix components, enabling their diffusion into the deeper regions of biofilm [6]. Importantly, it has been demonstrated that liposomes with neutral charge, such is the case of PEG-coated, can interact and fuse with bacterial cell envelope of Gram-positive and Gram-negative bacteria [7,8].

[1]          H. Le, C. Arnoult, E. Dé, D. Schapman, L. Galas, D. Le Cerf, C. Karakasyan, Antibody-Conjugated Nanocarriers for Targeted Antibiotic Delivery: Application in the Treatment of Bacterial Biofilms, Biomacromolecules. 22 (2021) 1639–1653. https://doi.org/10.1021/acs.biomac.1c00082.

[2]          J. Du, H.M.H.N. Bandara, P. Du, H. Huang, K. Hoang, D. Nguyen, S.V. Mogarala, H.D.C. Smyth, Improved Biofilm Antimicrobial Activity of Polyethylene Glycol Conjugated Tobramycin Compared to Tobramycin in Pseudomonas aeruginosa Biofilms, Mol. Pharm. 12 (2015) 1544–1553. https://doi.org/10.1021/mp500846u.

[3]          Y. Liu, H.J. Busscher, B. Zhao, Y. Li, Z. Zhang, H.C. van der Mei, Y. Ren, L. Shi, Surface-Adaptive, Antimicrobially Loaded, Micellar Nanocarriers with Enhanced Penetration and Killing Efficiency in Staphylococcal Biofilms, ACS Nano. 10 (2016) 4779–4789. https://doi.org/10.1021/acsnano.6b01370.

[4]          K. Forier, A.-S. Messiaen, K. Raemdonck, H. Deschout, J. Rejman, F. De Baets, H. Nelis, S.C. De Smedt, J. Demeester, T. Coenye, K. Braeckmans, Transport of nanoparticles in cystic fibrosis sputum and bacterial biofilms by single-particle tracking microscopy, Nanomedicine. 8 (2013) 935–949. https://doi.org/10.2217/nnm.12.129.

[5]          K. Forier, A.-S. Messiaen, K. Raemdonck, H. Nelis, S. De Smedt, J. Demeester, T. Coenye, K. Braeckmans, Probing the size limit for nanomedicine penetration into Burkholderia multivorans and Pseudomonas aeruginosa biofilms, J. Control. Release. 195 (2014) 21–28. https://doi.org/10.1016/j.jconrel.2014.07.061.

[6]          T.F. Bahamondez-Canas, H. Zhang, F. Tewes, J. Leal, H.D.C. Smyth, PEGylation of Tobramycin Improves Mucus Penetration and Antimicrobial Activity against Pseudomonas aeruginosa Biofilms in Vitro, Mol. Pharm. 15 (2018) 1643–1652. https://doi.org/10.1021/acs.molpharmaceut.8b00011.

[7]          S. Pereira, R.S. Santos, L. Moreira, N. Guimarães, M. Gomes, H. Zhang, K. Remaut, K. Braeckmans, S. De Smedt, N.F. Azevedo, Lipoplexes to Deliver Oligonucleotides in Gram-Positive and Gram-Negative Bacteria: Towards Treatment of Blood Infections, Pharmaceutics. 13 (2021) 989. https://doi.org/10.3390/pharmaceutics13070989.

[8]          L. Moreira, N.M. Guimarães, S. Pereira, R.S. Santos, J.A. Loureiro, M.C. Pereira, N.F. Azevedo, Liposome Delivery of Nucleic Acids in Bacteria: Toward In Vivo Labeling of Human Microbiota, ACS Infect. Dis. 8 (2022) 1218–1230. https://doi.org/10.1021/acsinfecdis.1c00601.

Round 2

Reviewer 3 Report

Comments and Suggestions for Authors

The authors satisfactorily addressed the comments and improved the quality of data. So, the reviewer feel that the manuscript can be accepted for publication.

Reviewer 4 Report

Comments and Suggestions for Authors

The article can be published as is, only in the Supplementary Data a personal comment from one of the co-authors must be removed.